



# Water mass structure and the effect of subglacial discharge in Bowdoin Fjord, northwestern Greenland

Yoshihiko Ohashi[1], Shigeru Aoki[2], Yoshimasa Matsumura[3], Shin Sugiyama[2,4], Naoya Kanna[4], Daiki Sakakibara[4]

[1]Department of Ocean Sciences, Tokyo University of Marine Science and Technology, Tokyo, 108-8477, Japan

[2]Institute of Low Temperature Science, Hokkaido University, Sapporo, 060-0819, Japan

[3]Atmosphere and Ocean Research Institute, The University of Tokyo, Kashiwa, 277-8564, Japan

[4]Arctic Research Center, Hokkaido University, Sapporo, 001-0021, Japan

*Correspondence to*: Yoshihiko Ohashi (yohash0@kaiyodai.ac.jp)

**Abstract.** Subglacial discharge has significant impacts on water circulation, material transport, and biological productivity in proglacial fjords of Greenland. To help clarify the fjord water properties and the effect of subglacial discharge, we investigated the water mass structures of Bowdoin Fjord in northwestern Greenland based on summer hydrographic observations, including turbidity, in 2014 and 2016. We estimated the fraction of subglacial discharge from the observational data and interpreted the observed differences in subglacial discharge behavior between two summer seasons with the numerical model results. At a depth of 60–80 m, temperature profiles were distinctively different in 2014 and 2016, and a larger fraction of submarine meltwater was detected in 2014. At a depth of 15–40 m, where the most turbid water was observed, the maximum subglacial discharge fractions near the ice front were estimated to be ~6 % in 2014 and ~4 % in 2016. The higher discharge fraction in 2014 was due to the stronger stratification, as suggested by numerical experiments performed with different initial stratifications. Turbidity near the surface was higher in 2016 than in 2014, suggesting a stronger influence of turbid subglacial discharge. The higher turbidity in 2016 could primarily be attributed to a greater amount of subglacial discharge, as inferred from the numerical experiments forced by different amounts of discharge. This study indicates that ambient fjord stratification difference is an important factor controlling the vertical distribution of subglacial discharge, together with its amount.

## 1 Introduction

In recent decades, the rate of ice mass loss from the Greenland ice sheet has increased from $51 \pm 65$ Gt a$^{-1}$ in 1992–2000 to $211 \pm 37$ Gt a$^{-1}$ in 2000–2011 (Shepherd et al., 2012). The acceleration of ice mass loss has been driven by increased surface melting and ice discharge from marine-terminating outlet glaciers (iceberg calving and submarine melting of glacier) (e.g., Andersen et al., 2015; Sasgen et al., 2012). Surface melt-induced meltwater discharge has accounted for 84% of the increase in ice mass loss since 2009, representing a more dominant contribution than ice discharge (Enderlin et al., 2014). Meltwater discharge has increased in response to warming air temperature in recent years (Fettweis et al., 2013a, 2013b; Hanna et al., 2008; Schrama et al., 2014).

Meltwater discharge from marine-terminating glaciers significantly affects fjord water circulation, material transport, and biological productivity (e.g., Carroll et al., 2015, 2017; Chu, 2014; Lydersen et al., 2014). Meltwater produced at the ice surface drains through crevasses to the base of the glacier. The submerged meltwater entrains sediments from a subglacial drainage system, gaining high turbidity. Once it flows out from the subglacial conduit, the turbid subglacial discharge forms an upwelling plume, entraining the ambient fjord water (Jenkins, 1999, 2011; Sciascia et al., 2013; Xu et al., 2013). Observations near Greenlandic glaciers have revealed that plume surface waters consist of 7–10 % subglacial discharge and ~90 % entrained fjord waters (Bendtsen et al., 2015; Mankoff et al., 2016), indicating that subglacial discharge plumes transport significant amounts of ambient deep water to the fjord surface. After upwelling, a substantial fraction of the subglacial discharge plume submerges and extends offshore at the lower part of the warm, fresh surface water (SW; Chauché et al., 2014; Xu et al., 2013).




A turbid water layer is observed at the subsurface where subglacial discharge spreads after the upwelling (Chauché et al., 2014; Stevens et al., 2016). Because the ambient deep water delivered by the plume is rich in nutrients, subglacial plume formation can enhance marine biological productivity (Arendt et al., 2011; Cape et al., 2019; Kanna et al., 2018; Lydersen et al., 2014; Meire et al., 2017). Conversely, high concentrations of suspended sediments near the fjord surface might reduce light

availability (Retamal et al., 2008). Therefore, it is important to understand the detailed behavior of subglacial discharge under various conditions of ambient water properties. However, only a few studies have described the temporal variations of the distribution of subglacial discharge and variables such as suspended sediment.

In situ observations have suggested that changes in the amount of subglacial discharge control the subglacial discharge distribution into a fjord (Chauché et al., 2014). Subglacial discharge is an intermittent process (e.g., Bartholomaus, et al., 2015;

Gimbert, et al., 2016). Although change in the amount of subglacial discharge can be an important controlling factor, the realistic influence on the subglacial discharge distribution has not been assessed in detail.

In addition to the amount of subglacial discharge, fjord stratification affects the behavior of subglacial discharge (e.g., Carroll et al., 2015). Warm, salty water of Atlantic origin (Atlantic Water: AW; e.g., Chauché et al., 2014; Straneo et al., 2012) occupies the deepest parts of Greenlandic fjords. The oceanic heat from AW can induce freshwater supply via melting of the

submarine glacier front, producing submarine meltwater (e.g., Straneo and Heimbach, 2013). Cold and relatively fresh water of Arctic origin (Polar Water: PW; e.g., Chauché et al., 2014; Myers et al., 2007; Ribergaard, 2007; Sutherland and Pickart, 2008), carried by the East and West Greenland Currents, overlies the AW layer. The properties of the water masses can change on various time scales from intra-seasonal to interannual or longer-term, reflecting the variabilities on much larger spatial scales. Near the surface of proglacial fjords, significantly warm, fresh and hence low density SW prevails, with properties that

are strongly affected by solar insolation, subglacial discharge, and iceberg and sea ice melts (Chauché et al., 2014; Mortensen et al., 2011). Therefore, fjord stratification, determined by fjord water properties and density, vary temporarily to reflect the amount of freshwater discharge and ambient water conditions. However, the impact of observed fjord stratification difference on the subglacial discharge distribution remains poorly understood.

To better understand structures of water properties in a fjord under the influence of subglacial discharge, we performed

summer hydrographic and turbidity observations in Bowdoin Fjord (BF) in northwestern Greenland. We used the observational data to estimate the fraction of subglacial discharge in fjord water. Results obtained in 2014 and 2016 were compared with those computed with a subglacial plume model. In the northwestern sector of the Greenland ice sheet, ice mass loss has increased since 2005 (Khan et al., 2010; Kjær et al., 2012). Further changes are expected in the future, but in situ data are relatively sparse, and the detailed fjord water structure and the effect of subglacial discharge are poorly understood in this

region. We chose the BF in northwestern Greenland as the study area because of its proximity to a village as well as field observations conducted on glaciers and the ocean in the area (Kanna et al., 2018; Sugiyama et al., 2015). Such previous and ongoing studies in the region provided the impetus to conduct the study reported in this paper.

The focus of our measurements in BF was to reveal fjord water properties and the differences in water structures in 2014 and 2016. The aim of the study is to determine the factors controlling changes in the water mass structure of proglacial fjord.

This study is structured as follows. We introduce the target area as the BF in northwestern Greenland (Sect. 2). We explain the details of conductivity–temperature–depth (CTD)/turbidity observations, freshwater fraction analysis, and numerical experiments (Sect. 3). Next, we present the differences in fjord water properties and subglacial discharge distribution between 2014 and 2016 (Sect. 4). Finally, we compare the observational data with model results to interpret the mechanism controlling the differences in the fjord water properties (Sect. 5).


## 2 Study area

This study focused on BF (77.6° N, 66.8° W; 3–5 km wide and 20 km long), one of the arms of Inglefield Bredning (IB; 10–15 km wide and 100 km long) in northwestern Greenland (Fig. 1). In IB, sea ice covers the ocean between October and May.





In June and July, sea ice melts rapidly, and open ocean surface appears. During the summer melt season, areas of highly turbid ocean surface water form near the ice sheet and glaciers as a result of glacial meltwater discharge (Ohashi et al., 2016).

The bathymetry of BF was surveyed with an echo sounder along the centerline and several profiles across the fjord (Sugiyama et al., 2015). The water depth is about 600 m at the mouth of BF, shoaling to about 210 m at the ice front of the

Bowdoin Glacier (BG). Hence, it is assumed that subglacial discharge takes place at a depth of 210 m below sea level, which corresponds to the depth between warm AW (at the deepest part of the fjord) and cold PW (depth: 50–150 m). Water properties at this depth are expected to change according to the relative influence of AW and PW. Therefore, BF is suitable for assessing the impact of the change in water properties on the distribution of subglacial discharge.

To help estimate the subglacial discharge conditions, air temperature data taken at Qaanaaq Airport located ~30 km

southwest of the BG were used (77.47° N, 69.23° W, 16 m a.s.l.; blue circle in Fig. 1a). Spring–summer air temperature was generally warmer in 2016 than in 2014. The amount of subglacial discharge is controlled by the amount of surface melt, which is commonly estimated as the sum of the positive degree days (PDD) (e.g., Cuffey and Paterson, 2010). The PDD at Qaanaaq Airport was approximately 20 % greater in 2016 than in 2014, suggesting a greater subglacial discharge in 2016. In addition, this PDD difference means that at least 20 % difference in the amount of discharge between two observations could occur,

assuming that the amount of subglacial discharge increases at a constant rate during the summer melt season.

## 3 Data and methods

### 3.1 Hydrographic observation

We performed CTD observations in BF in the summers of 2014 and 2016. The observations were performed along the

centerline of the fjord at six locations each year; at Stations 14D1–6 on 4 August 2014, and at Stations 16D1–6 on 29 July 2016 (where the first two digits in the station label denote the year; Fig. 1b). Station 16D6 was located in IB, approximately 4 km from the mouth of BF. A CTD profiler (RINKO Profiler ASTD-102, JFE Advantech) was lowered from a boat to measure the temperature, salinity, and turbidity profiles from the surface to the bottom of the fjord (see Kanna et al., 2018 for details). The precision of the depth, temperature, salinity, and turbidity measurements were 1.8 m, 0.01 °C, 0.01, and 0.3 formazin

turbidity units (FTU), respectively.

We collected 33 water samples at Stations 16D2–6 to calibrate the salinity measurements. Water sampling was performed at depths deeper than 10 m to avoid the influence of the steep salinity gradient near the surface. The salinities of the water samples were measured using a salinometer (Guideline Autosal 8400B) to correct the in situ measurements based on the CTD. The uncertainty in salinity (~0.01) made it difficult to compare its absolute value, but the vertical gradient of salinity should

be valid.

### 3.2 Freshwater fraction analysis

In proglacial fjords, seawater is influenced by freshwaters from subglacial meltwater discharge (subglacial discharge) and submarine melting of the ice front (submarine meltwater). A potential temperature–salinity ($\theta$–$S$) diagram can be used to

separate the mixing processes of these waters (see also Appendix A).

Subglacial discharge mixed with the ambient ocean water to form an upwelling plume, which subsequently spread. The straight line on the $\theta$–$S$ diagram between the subglacial discharge ($\theta = \theta_{sg}= 0$ °C, $S = S_{sg}= 0$) and ambient ocean water at the conduit depth ($\theta = \theta_e$, $S = S_e$; potential temperature and salinity at the 210 m depth averaged for all observation sites in each year; hereinafter referred to as "entrained fjord water") is called the runoff line (Straneo et al., 2011, 2012).

At the ice front, submarine melting of ice is driven by the heat of ambient seawater. The straight line on the $\theta$–$S$ diagram that indicates the mixing caused by submarine melting is called the melt line (Straneo et al., 2011, 2012). We defined the effective potential temperature ($\theta_{mw}$: °C) by calculating the energy required to melt ice when $S = 0$ (Chauché et al., 2014; Gade, 1979; Jenkins, 1999; Straneo et al., 2012):




$$\theta_{\mathrm{mw}} = \theta_{\mathrm{f}} - \frac{L_{\mathrm{f}}}{c_{\mathrm{p}}} - \frac{c_{\mathrm{i}}(\theta_{\mathrm{f}} - \theta_{\mathrm{i}})}{c_{\mathrm{p}}} \quad (^{\circ}\mathrm{C}), \tag{1}$$

where $\theta_{\mathrm{f}}$ is the pressure-corrected melting point of ice ($-0.1$ °C), $L_{\mathrm{f}}$ is the latent heat of fusion (334.5 kJ kg$^{-1}$), $\theta_{\mathrm{i}}$ is ice temperature ($-5$ °C; Seguinot et al., 2016), and $c_{\mathrm{i}}$ and $c_{\mathrm{p}}$ are the specific heat capacities of ice and seawater (2.1 and 3.98 kJ kg$^{-1}$ K$^{-1}$). Thus, the melt line is the line connecting the submarine meltwater ($\theta = \theta_{\mathrm{mw}}$, $S = S_{\mathrm{mw}} = 0$) and the entrained fjord

water.

In this study, the $\theta$–$S$ data of entrained fjord water were located between the AW and the PW cores, suggesting that the water property was influenced by the mixing of AW and PW. Moreover, the characteristics of AW core differed between 2014 and 2016. Hence, the entrained fjord water property used in this study differed between 2014 and 2016. This entrained fjord water is not necessarily the sole endmember of submarine meltwater. However, since the observed temperature has a similar

structure at the depths where the submarine melt is in effect, we selected the entrained fjord water as the representative endmember of submarine meltwater fraction (see Sects. 4.2 and 5.3 for details). Note that the estimation of the subglacial discharge fraction is little affected by the above setting of the endmember due to the $\theta$–$S$ inclination proximity to the melt line.

Assuming that the water properties can be described as a mixture of the three different water masses (subglacial discharge, submarine meltwater, and entrained fjord water), the fraction of each water component in seawater can be calculated (Appendix

A; e.g., Everett et al., 2019; Mankoff et al., 2016; Mortensen et al., 2013). In the $\theta$–$S$ space consisting of the positive fraction of each component (hereinafter referred to as the "meltwater quadrant"), the water mass properties can be explained as a mixture of the three components. Note that water mass properties outside the meltwater quadrant are affected by other mixing processes, and the calculation mentioned above is not applicable.

**3.3 Numerical experiment**

To interpret the observed differences in the subglacial discharge distribution between 2014 and 2016, we perform a set of numerical model experiments. The model simulates a transient behavior of the subglacial discharge plume in front of the BG (Fig. 2). We use a three-dimensional non-hydrostatic ocean model with the Boussinesq approximation, originally developed by Matsumura and Hasumi (2008). The model domain represents BF and is 3.2 km wide (from east to west; x-direction), 20.5

km long (from north to south; y-direction), and 600 m deep (z-direction) (Figs. 2a and 2b). The ice front is located at the northern end and the fjord mouth at the southern end. We simplify the measured bathymetry of the BF (Sugiyama et al., 2015; Figs. 2a and 2b) to set a model geometry. A subglacial drainage conduit is approximated by a rectangular channel (200 m wide $\times$ 50 m high) at the base (210 m deep) and the center of the glacier front. The model resolution is 50 m horizontally and 10 m vertically. The horizontal subgrid-scale viscosity and diffusion are represented by the strain rate-dependent Smagorinsky

model (Smagorinsky, 1963) following Matsumura and Hasumi (2010). The vertical viscosity and diffusivity coefficient are set to $1.0 \times 10^{-5}$ m$^2$ s$^{-1}$. The Coriolis parameter is set to $1.4 \times 10^{-4}$ s$^{-1}$.

The initial potential temperature and salinity are set to be horizontally uniform using the observation data in 2014 (solid lines in Figs. 2c and 2d). Subglacial discharge ($\theta = 0$ °C, $S = 0$) is injected into the model domain from the subglacial drainage conduit at the northern boundary. The velocity profile at the southern boundary is predicted to compensate for the discharge

inflow so that the total water volume is conserved in the model domain. A virtual tracer, that is assumed to obey the same advection–diffusion equation as potential temperature and salinity, is implemented to track the behavior of subglacial discharge. The tracer concentration is initially zero over the whole domain and exhibits unity for the subglacial discharge. No heat flux and wind stress are applied at the surface. No-slip conditions are used for the seafloor. We restore $\theta$ and $S$ to the initial profile at the southern boundary. The model is integrated for five days from a state of rest.

As the control case, we perform the experiment given by inflow velocity at the northern boundary ($V_{\mathrm{sg}}$) of 0.05 m s$^{-1}$, corresponding to the subglacial discharge ($Q_{\mathrm{sg}}$) of 500 m$^3$ s$^{-1}$ (hereinafter referred to as CTRL; Tab. 1). To investigate the effect of the amount of subglacial discharge, the experiments are conducted by changing the inflow velocity by a factor of ten





($V_{sg}$ = 0.01, 0.06, and 0.1 m s$^{-1}$; $Q_{sg}$ = 100, 600, and 1000 m$^3$ s$^{-1}$; hereinafter referred to as Q100, Q600, and Q1000, respectively). We also perform the experiment with the same subglacial discharge as the CTRL but with the initial stratification as observed in 2016 to assess the influence of the stratification difference (hereinafter referred to as ST16; dashed lines in Figs. 2c and 2d).

Although we have implemented the effect of submarine melting following Holland and Jenkins (1999) and Losch (2008), the amount of resultant meltwater is less than 0.05 % of the imposed subglacial discharge. Hence, the effect of submarine melting is negligible on the time scale considered with the present model settings.

## 4 Results

### 4.1 Water properties

From the bottom to the surface, layers of warm, saline AW (its core represented by potential temperature maximum; $\theta_{max}$), cold PW (its core represented by the potential temperature minimum; $\theta_{min}$), and significantly warm, fresh SW were observed in 2014 and 2016 (Fig. 3). In 2016, the $\theta$–$S$ property was similar at depths deeper than the PW core inside and outside BF. However, at depths shallower than the PW core, the $\theta$–$S$ properties differed. Outside BF, temperature monotonically increased

from the PW core toward the surface, suggesting the development of a seasonal pycnocline. Inside BF, temperature increased upwards but then decreased at about 40 m. This difference was suggestive of local characteristics of the fjord water structure. Between 2014 and 2016, there were several differences in water properties, especially in temperature. The vertical distributions of the potential temperature, salinity, and turbidity and their differences are described in the subsequent sections.

### 4.1.1 Potential temperature

The vertical structures of temperature were notably different between 2014 and 2016. The AW core was warmer and shallower in 2014 (1.3 °C at ~290 m) than in 2016 (1.0 °C at ~320 m; Figs. 4a–4d). At the deepest part of the fjord, the warm layer (> 0 °C) was thicker in 2014. Moreover, the depth of the PW core was shallower in 2014 with a temperature of −0.8 °C in both observations. Because of the differences in the AW core temperature and warm layer thickness, the temperature near the BG

drainage conduit (210 m) was up to 0.9 °C warmer in 2014 (Fig. 4e).

At depths shallower than 150–170 m (i.e., PW core), the temperature structure differed significantly between the two observations. In 2014, a relatively cold temperature maximum (−0.7 °C; hereinafter referred to as "local $\theta_{max2014}$") was found at a depth of 100 m (Figs. 3, 4a, and 4b). Moreover, the coldest water (−1.0 °C; hereinafter referred to as "local $\theta_{min}$") was observed at a depth of 80 m (Figs. 3, 4a, and 4b), and the temperature at the local $\theta_{min}$ was even colder than that at the PW core.

In 2016, a corresponding local $\theta_{min}$ was not observed, but a clear temperature maximum (0.2 °C; hereinafter referred to as "local $\theta_{max2016}$") was found at a depth of 60 m and the temperature of the local $\theta_{max2016}$ was significantly warmer than that of the local $\theta_{max2014}$ (Figs. 3, 4c, and 4d). Because the increase in temperature from the PW core to the local $\theta_{max2016}$ in BF was roughly the same as that outside BF, the water properties below the local $\theta_{max2016}$ layer in the fjord could represent a seasonal pycnocline over a wider area. In addition, the difference between local $\theta_{max2016}$ and local $\theta_{max2014}$ suggested that more enhanced

seasonal pycnocline was developed in 2016 than in 2014.

At depths shallower than 20 m, the water temperature increased rapidly toward the surface in both 2014 and 2016. However, the temperatures were up to 2.3 °C colder in 2016 than in 2014 at depths of 5–20 m (Figs. 4e and 4f).

### 4.1.2 Salinity, potential density, and stratification

Salinity varied vertically between 2014 and 2016. At depths below 210 m, including the AW core, salinity was roughly the same between the two observations (Figs. 5a–5d), although the salinity at the AW core differed (2014: 34.1, 2016: 34.2). At a depth of 5–170 m (shallower than the PW core), salinity was higher in 2016 than in 2014, including the salinity at the PW core



(2014: 34.1, 2016: 34.2) (Figs. 5e and 5f). This difference was more significant (0.6–1.6) near the surface (5–20 m). In contrast, salinity at the surface (0–5 m) was lower in 2016 except at the outer portion of BF.

The vertical distribution of the potential density was mainly controlled by salinity; therefore, the differences in the potential density between 2014 and 2016 were mostly the same as those of salinity (not shown). At a depth of 5–170 m, the potential

density was higher in 2016 than in 2014, while at the surface it was lower in 2016.

Behavior of subglacial discharge is affected by salinity and density profiles. The square of the Brunt-Väisälä frequency ($N^2$) increased toward the surface in both observations. In particular, $N^2$ was the greatest ($> 0.001$ s$^{-2}$) at depths shallower than 10–15 m, representing the strongest stratification among all depths (Figs. 6a–6d). $N^2$ was higher in 2016 than in 2014 at depths shallower than 10 m, but lower by up to 0.0007 s$^{-2}$ at depths of 10–50 m (Figs. 6e and 6f). Thus, the stratification in 2016 was

stronger near the surface (0–10 m) than in 2014 but weaker at the subsurface (10–50 m).

### 4.1.3 Turbidity

Turbidity acts as an effective tracer of subglacial discharge. The highest turbidity layer ($> 4$ FTU) was found not at the surface, but at the subsurface at 15–50 m in 2014 and 10–40 m in 2016 (Figs. 7a–d). Turbidity decreased from the subsurface to a depth

of about 100 m, and reached almost zero at depths below 150 m. In addition, turbidity decreased with distance from the ice front toward the mouth of the fjord, in contrast to temperature and salinity, which changed little horizontally.

The distribution in turbidity differed between the two observations. In 2014, a low turbidity layer existed further offshore at a depth of around 10 m, which was sandwiched by higher turbidities above and below (Figs. 7a and 7b). Conversely, in 2016, turbidity was nearly homogeneous at the depth range of 0–15 m, and there was no discontinuity at a depth of 10 m (Figs. 7c

and 7d). Therefore, turbidity at a depth of 0–15 m was 1–2 FTU higher in 2016 than in 2014 (Figs. 7e and 7f). Meanwhile, at the depth of 40–150 m within 5 km of the ice front in 2014, turbidity was relatively high ($> 1$ FTU). In 2016, no relatively high turbidity layer existed at depths below around 60 m. Therefore, turbidity at a depth of 15–200 m within 5 km of the ice front was lower in 2016 than in 2014. This difference was particularly significant (up to 1.5–5 FTU) at depths of 20–120 m. These differences in turbidity could be attributed to the fraction of subglacial discharge, as will be discussed in Sect. 4.2.

### 4.2 Freshwater fraction in $\theta$–$S$ diagram and the role of subglacial discharge

As shown in Sect. 4.1, there were some differences in the vertical distribution of the $\theta$–$S$ properties between 2014 and 2016. To understand the differences in the mixing processes controlling the water properties, we estimated the freshwater fractions in the $\theta$–$S$ diagram (see Sect. 3.2 for details). We compared a common site nearest the ice front (Stations 14D1 and 16D3;

approximately 4 km from the ice front) and examined the difference in the freshwater fractions. The results were similar for the other stations.

First, we examined the properties of the entrained fjord water at the 210 m depth. The $\theta$–$S$ properties obtained in 2014 were closer to those of the AW core than the PW core (Figs. 8a and 8c), whereas they were more similar to the PW core (Figs. 8b and 8d). This difference implies greater influence of AW at the 210 m depth in 2014.

At depths from 210 m to the PW core (~150 m), the $\theta$–$S$ properties in 2014 showed a similar tendency as those in 2016. In both 2014 and 2016, near the PW core, the $\theta$–$S$ properties deviated slightly from the melt line and were located outside the meltwater quadrant, implying the influence of PW.

The $\theta$–$S$ properties above the PW core (80–150 m) differed between 2014 and 2016. The $\theta$–$S$ properties in 2014 deviated slightly toward a high temperature from the PW core to the local $\theta_{max2014}$ (~100 m). This deviation might reflect a slightly

developed seasonal pycnocline. From the local $\theta_{max2014}$ (~100 m) to the local $\theta_{min}$ (~80 m), the $\theta$–$S$ properties aligned perfectly along the melt line (solid black lines in Figs. 8a and 8c). The submarine meltwater fraction at the local $\theta_{min}$ was estimated to be 1.6 %, which was the largest fraction, indicating the greatest influence of submarine melting among all depths (Fig. 8c). The estimated submarine meltwater fraction at the local $\theta_{min}$ increased to 2.2 % and 2.5 % in the case that the endmember of





entrained fjord water was set to the water at the depth of 250 m and 300 m, respectively. Meanwhile, the subglacial discharge fraction was not significant. By contrast, in 2016, the $\theta$–$S$ properties were indicative of a seasonal pycnocline above the PW core, and the submarine meltwater fraction decreased to less than 0.5 % (Fig. 8d). The estimated fraction increased to 1.1 % and 1.8 % in the case that the endmember of entrained fjord water was set to the water at the depth of 250 m and 300 m, respectively, but the submarine meltwater fraction was by far smaller than those in 2014. On the other hand, the subglacial discharge fraction was estimated to be 1.1 %.

At depths of 50–80 m, above the local $\theta_{min}$ in 2014, the seawater consisted of 1.3–1.6 % submarine meltwater, 0.1–1.4 % subglacial discharge, and 97.3–98.3 % entrained fjord water (Fig. 8c and blue lines in Fig. 9). This mixture reflected the substantial influence of submarine meltwater in this layer. In 2016, the $\theta$–$S$ data were located outside the meltwater quadrant, implying that the ocean water properties could not be explained by the simple mixing of the three water components (Figs. 8b and 8d). Near the local $\theta_{max2016}$ approximately 60 m, turbidity was significantly lower in 2016 than in 2014, implying a weaker influence of subglacial discharge.

Further upward, we focused on the subsurface at a depth of 15–40 m where the highest turbidity was observed. The subglacial discharge fraction was high, with a maximum around 15 m (Figs. 8c and 8d). In 2014, the seawater consisted of 2.5–6.0 % subglacial discharge, 0.4–1.1 % submarine meltwater, and 93.6–96.3 % entrained fjord water (Fig. 8c and blue lines in Fig. 9). Although the submarine meltwater fraction decreased closer to 15 m, the rapid increase in temperature in the $\theta$–$S$ diagram might reflect the influence of the development of a seasonal pycnocline. In 2016, the seawater was composed of approximately 2.4–4.0 % subglacial discharge and 96.0–97.6 % entrained fjord water, with no submarine meltwater (Fig. 8d and red lines in Fig. 9). The subglacial discharge fraction was up to 2.0 % greater in 2014 than in 2016.

Near the surface (depth: 5–15 m) immediately above the most turbid water layer, the $\theta$–$S$ properties were outside the meltwater quadrant in both observations. The $\theta$–$S$ properties in 2014 deviated toward a significantly high temperature and low salinity above the runoff line (Figs. 8a and 8c). The $\theta$–$S$ properties in 2016 showed a similar tendency to 2014, but the deviation from the runoff line was smaller (Figs. 8b and 8d). Therefore, water might have been influenced by subglacial discharge more strongly in 2016 than in 2014, although it was difficult to quantify, because this layer was outside the meltwater quadrant. Turbidity at this depth was also higher in 2016 than in 2014, supporting the greater influence of subglacial discharge in 2016.

## 5 Discussion

### 5.1 Quantitative relationship between the subglacial discharge fraction and turbidity

In Sect. 4.2, high turbidity corresponded to a high fraction of subglacial discharge. Therefore, we assessed the quantitative relationship between turbidity and subglacial discharge fraction in both study years (Fig. 10). In 2014, the relationship between the subglacial discharge fraction ($R_{sg}$: %) and turbidity ($TUR$: FTU) in the meltwater quadrant was expressed as $R_{sg} = TUR \times 0.7 - 2.0$ ($R^2 = 0.67$; Fig. 10a). In 2016, the data in the meltwater quadrant and that including the data points close to the runoff line (depth: 15–40 m) showed a linear relationship of $R_{sg} = TUR \times 0.6 + 0.3$ ($R^2 = 0.94$), with a roughly similar inclination to that in 2014 (Fig. 10b). Moreover, the low turbidity at the local $\theta_{max2016}$ (depth: 60 m) in 2016 was consistent with the calculation that the fraction of subglacial discharge is small in this layer (Figs. 8b and 8d). These results indicate that the vertical distribution of turbidity reflects the mixing ratio of subglacial discharge near the ice front (Fig. 7).

Recent observations in other regions of Greenland have qualitatively shown that the high turbidity subsurface layer corresponds to the distribution of subglacial discharge (Chauché et al., 2014; Stevens et al., 2016). The quantitative relationship between turbidity and the subglacial discharge fraction presented in this study reveals that measuring turbidity is an effective tool to investigate the distribution of subglacial discharge into fjords. At the fjord surface, the quantitative relationship between turbidity and the subglacial discharge fraction was not investigated in this study. However, because the turbidity distribution at the fjord surface can be visually captured by satellites (Ohashi et al., 2016) and drones, it is easier to monitor the distribution of turbidity than to investigate the subglacial discharge distribution based on in situ observations. This emphasizes the



importance of turbidity measurements to better understand subglacial discharge distribution.

**5.2 Factors controlling the observed subglacial discharge distribution**

As shown in Sect. 4.2, the estimated subglacial discharge fraction differed between 2014 and 2016. A likely interpretation of
this difference is the amount of subglacial discharge and fjord stratification in each year. To test this hypothesis, we compare
the numerical model results with the observational data.

Numerical experiments are performed to investigate the impacts of a 20 % greater discharge (Q600) and fjord stratification
difference (ST16) as compared to the most realistic case (CTRL; see Appendix B for details) (Fig. 11). The 20 % greater
amount of discharge was assumed in Q600 based on 20% increase in PDD at Qaanaaq Airport from 2014 to 2016 (see Sect.
2). When comparing the results of CTRL with those of Q600 and ST16, the results after 15 h are used to consider the integration
time required for the virtual tracer to reach the southern boundary in the earliest case (16 h in CTRL, 15 h in Q600, and 15 h
in ST16).

Near the fjord surface (depth: 5–15 m), turbidity was higher in 2016 than in 2014 (Figs. 7e and 7f). In Q600, the
concentration of subglacial discharge tracer near the surface (depth: 0–20 m) increased by 10–40 % from that in CTRL (Figs.
11a and 11c). The observation and model were consistent with the assumption that the subglacial discharge in 2016 was greater,
as inferred by the PDD. These results suggest that turbid subglacial discharge extended near the fjord surface more in 2016
because of a stronger buoyancy forcing exerted by the 20 % greater discharge. Chauché et al. (2014) compared the $\theta$–$S$
properties of fjord water with the amount of subglacial discharge based on a glacier surface melt estimation using a PDD/melt-
rate model (Box, 2013). Comparison of results in CTRL and Q600 showed that a 20 % change in the subglacial discharge
caused an approximately 10–40 % change in the subglacial discharge fraction near the surface. In ST16, the tracer
concentration increased at a depth of 0–10 m and decreased at a depth of 10–20 m from that in CTRL (Figs. 11b and 11d).
Although the increase at 0–10 m favored the observed change, the mean increase in the tracer concentration was smaller than
that of the increase in discharge. Therefore, the distribution of subglacial discharge near the surface could be affected more by
a change in the amount of subglacial discharge than the difference in the observed fjord surface stratification.

When the amount of subglacial discharge is large, highly turbid glacial meltwater is expected to spread over a larger surface
area. This is consistent with remote sensing data analyses performed off the coast of northwestern Greenland where a number
of glaciers terminate in the ocean (Ohashi et al., 2016). Furthermore, a 20 % increase in the subglacial discharge results in a
greater tracer concentration not only at the surface but also at the subsurface (depth: 20–40 m). The magnitude of the change
was similar between the surface and subsurface layers (a few tens of percent) (Figs. 11a and 11c). This implies the need to
consider the vertical distribution of subglacial discharge at the subsurface in addition to the satellite surface measurements to
quantitatively assess the overall impact of subglacial discharge.

In contrast to the observations near the surface, the fraction of turbid subglacial discharge at the subsurface (15–40 m) was
greater in 2014 than in 2016 (Figs. 7, 8, and 9a). This field observation was inconsistent with the numerical experiments of the
differences in discharge, showing a smaller concentration of subglacial discharge tracer at a depth of 20–40 m under a smaller
amount of discharge (Figs. 11a and 11c). From the stratification change experiment, the tracer concentration at the subsurface
was about 10–20 % greater in CTRL than in ST16 (Figs. 11b and 11d). This was consistent with the observed difference at the
subsurface, and the rate of change was quantitatively consistent (a few tens of percent). Thus, variations in stratification are a
likely explanation for the observed differences in the subglacial discharge fraction at the subsurface.

Previous studies have shown that strong subsurface stratification in fjords prohibits upwelling of the subglacial discharge
plume and results in the dispersion of discharge into a subsurface layer (Carroll et al., 2015). Furthermore, plumes extend
further over the surface under weaker stratification (Carroll et al., 2015). The stratification in ST16 in the subsurface layer is
weaker than that in CTRL, whereas that near the surface is stronger. After reaching the surface, the plume in ST16 is less likely
to submerge and spreads near the surface in higher concentrations than that in CTRL (Figs. 11b and 11d). In the case of when



the plume reached the fjord surface, our model suggested that strong surface stratification would prohibit the subduction of the outcropped plume, likely resulting in a plume that extends to the fjord surface.

### 5.3 Difference in the formation process of stratified structures

Subsurface stratification influences the distribution of subglacial discharge. The fjord stratification in 2014 and in 2016 differed at a depth of approximately 60–80 m which was attributed to the influence of submarine melting and the broad seasonal pycnocline. In this section, we discuss the formation process of the stratified structures in each observation.

In 2014, a warm layer strongly affected by AW was in contact with the ice front, which could enhance the fraction of submarine meltwater around a depth of 80 m (Figs. 3, 4, 8a, 8c, and 9b). Because the submarine meltwater fraction was detected regardless of the distance from the ice front of the BG, submarine meltwater from other glaciers in IB might have influenced the water in BF. Moreover, this horizontal distribution of the submarine meltwater suggests that the submarine melt does not take place only at the subglacial conduit depth. However, the relationship that temperature at the deep part of fjord is generally warmer in 2014 and the relatively warmer endmember temperature of entrained fjord water in 2014 should be robust (Figs. 4 and 8). A warm layer above 210 m extended further up to the shallower layer in 2014 than in 2016. The available excess heat of up to 0.9 °C was 1.6 times greater in 2014 than in 2016 when calculated by the difference from the freezing temperature (Figs. 4 and 8). A previous model study found that the rate of submarine melting increased proportionally to the increase in water temperature to the power of 1.3–1.6 (Xu et al., 2013). Porter et al. (2014) revealed that the rates of ice mass loss at the Tracy and Heilprin Glaciers, neighboring tidewater glaciers in IB, differed substantially between 1.63 and 0.53 Gt a$^{-1}$. Since the water depth at the ice front of the Tracy Glacier (610 m) is deeper than that of the Heilprin Glacier (350 m), the ice front has wider contact with warm AW, suggesting a greater glacier mass loss associated with the greater submarine melting (Porter et al., 2014). Our study suggests that the difference in the structure of deep heat storage can alter the development of submarine melting layer and affects ice mass loss from Greenland glaciers.

In 2016, a simple, but broad seasonal pycnocline was detected outside BF (Figs. 3, 4, 8b and 8d). In the study area (Qaanaaq Airport; blue circle in Fig. 1a), the mean temperature during the previous winter (December–February) was 1.1 °C lower in 2016 than in 2014. Thus, winter vertical mixing of the fjord was enhanced and the mixed layer depth could deepen during the preceding winter in 2016. In summer, a seasonal pycnocline could develop above the remnant of the winter mixed layer. In addition, the PW core in BF was deeper in 2016, supporting the possibility of the development of a seasonal pycnocline influenced by the enhancement of the winter vertical mixing. We speculate that the development of the seasonal pycnocline over a broader area in 2016 is due to the enhancement of winter vertical mixing.

At depths shallower than 60–80 m, where subglacial discharge spreads, fjord stratification could be modified by subglacial discharge. In general, fjord stratification is expected to be stronger after subglacial discharges into the fjord, because a density difference is generated. However, this study revealed the transitional processes of subglacial discharge over a relatively short time scale. To fully understand the longer-term interactions of subglacial discharge and fjord stratification (e.g., seasonal and interannual variations), we need to perform long-term oceanic observations and numerical experiments to capture the realistic nature of discharge and submarine melting over a much broader model domain.

### 6 Conclusions

With a focus on the differences in the subglacial discharge distribution and ambient water properties, we investigated the water mass structures in BF in northwestern Greenland. The differences in the distribution of subglacial discharge and water mass structure between 2014 and 2016 are summarized in Fig. 12.

The depths of the temperature minimum and maximum differed between the two observations. The $\theta_{max}$ (AW core) and the $\theta_{min}$ (PW core) were shallower in 2014 ($\theta_{max}$ at ~290 m; $\theta_{min}$ at ~150 m) than in 2016 ($\theta_{max}$ at ~320 m; $\theta_{min}$ at ~170 m). The local $\theta_{min}$ was observed around 80 m in 2014 but not in 2016. In turn, prominent local $\theta_{max2016}$ was detected around 60 m in



2016, which was significantly warmer than that in 2014. The analysis of the $\theta$–$S$ properties indicated that local $\theta_{min}$ in 2014 and local $\theta_{max2016}$ in 2016 could be influenced by the development of a submarine melting layer and broad seasonal pycnocline, respectively.

Subglacial discharge spread at a depth shallower than local $\theta_{min}$ and local $\theta_{max2016}$. In both 2014 and 2016, the fractions of turbid subglacial discharge were highest at the subsurface (15–40 m). The maximum fraction near the ice front was ~6 % in 2014 and ~4 % in 2016. Near the surface (5–15 m), turbidity was higher in 2016 than in 2014, suggesting a stronger influence of turbid subglacial discharge in 2016.

To assess the factors controlling the difference in the observed subglacial discharge distribution, we perform a set of numerical experiments to simulate the subglacial discharge distribution under different stratification and volume flux of
subglacial discharge. The experiments using the different initial stratifications suggest that the fractional difference in subglacial discharge at the subsurface is attributed to the difference in fjord stratification. Moreover, the numerical model results based on a 20 % greater discharge suggest that the difference near the surface is primarily affected by the increase in discharge.

From the surface to the subsurface, where subglacial discharge spread, fjord stratification varied between the study years
depending on the layer that developed and the amount of subglacial discharge. At a depth around 60–80 m, fjord stratification could be determined by the influences of submarine melting and seasonal pycnocline. Because of the thicker warm layer strongly affected by AW, a submarine melting layer was able to develop in 2014.

Our study suggests that observed fjord stratification, together with the amount of subglacial discharge, can affect the distribution of subglacial discharge. Given the current increase in meltwater discharge from Greenlandic glaciers, the buoyancy
forcing of the subglacial discharge plume and ambient fjord stratification are expected to change. To fully capture the subglacial discharge distribution, further continuous observations and numerical modeling are required over a wider area encompassing northwestern Greenland.

**Appendix A: Freshwater fraction analysis equations**

The origins of freshwater in ocean water can be estimated based on a freshwater endmember analysis. To quantitatively assess the difference in freshwater fractions, we calculated the volume fractions of subglacial discharge ($f_{sg}$), submarine meltwater ($f_{mw}$) and entrained fjord water ($f_e$) from the observed temperature and salinity (Fig. A1). From the mass conservation of $f_{sg}$, $f_{mw}$, and $f_e$:

$$f_{sg} + f_{mw} + f_e = 1 . \tag{A1}$$

The sampled potential temperature ($\theta_A$: °C) and salinity ($S_A$) are represented in the following equations:

$$\theta_A = \theta_{sg}f_{sg} + \theta_{mw}f_{mw} + \theta_e f_e \quad \text{(°C)}, \tag{A2}$$

$$S_A = S_{sg}f_{sg} + S_{mw}f_{mw} + S_e f_e . \tag{A3}$$

Because $S_{sg} = 0$ and $S_{mw} = 0$, Eq. (A3) is converted into:

$$S_A = S_e f_e . \tag{A4}$$

Using Eqs. (A1), (A2), and (A4), $f_{sg}$ and $f_{mw}$ are given as the following respective equations:

$$f_{sg} = \frac{1}{\theta_{sg}-\theta_{mw}}\left(\theta_A - \theta_{mw}\left[1-\frac{S_A}{S_e}\right] - \theta_e \frac{S_A}{S_e}\right), \tag{A5}$$

$$f_{mw} = 1 - f_{sg} - \frac{S_A}{S_e} . \tag{A6}$$

**Appendix B: Validation of the transitional process of subglacial discharge in the numerical experiment**

To validate the transitional process of subglacial discharge obtained from the numerical experiment, we compare the numerical model results (CTRL, Q1000, and Q100) with observation condition.

In CTRL, the highest tracer concentration is observed at a depth of 0–10 m within 1 km from the ice front and at a depth of



20–30 m beyond 1 km from the ice front (Figs. B1a and B2). The distribution generally coincides with the observed profiles.

In Q1000, beyond a few kilometers from the ice front, the depth of the highest tracer concentration is observed at a depth of 20–30 m, approximately the same results as obtained in CTRL (Figs. B1b and B2). However, the region of highest tracer concentration covers the entire width of the fjord surface (Figs. B1e and B1h), which is not detected at the BG ice front during the two years of observations.

In Q100, highest concentration of subglacial discharge tracer is observed at a depth of 20–40 m regardless of the distance from the ice front (Figs. B1c and B2). This result is significantly different from those in CTRL and Q1000. The results indicate that subglacial discharge does not reach the fjord surface, which is inconsistent with the visual observation of turbid surface plumes in front of BG.

In summary, among the three experiments performed under the same initial stratification, the results of CTRL are the most consistent with the observed horizontal and vertical distributions of the subglacial discharge. It should be noted that the above analyses represent the transitional process until the arrival of the tracer at the southern boundary, and are not applicable to long-term behavior of subglacial discharge.

## Code availability

The source code for the non-hydrostatic ocean model used in this study is available at http://lmr.aori.u-tokyo.ac.jp/feog/ymatsu/kinaco.git.

## Data availability

Air temperature data are provided by the United States National Oceanic and Atmospheric Administration National Climatic Data Center (http://www.ncdc.noaa.gov/cdo-web/). The CTD data in 2014 are available from the first author upon request and those in 2016 from Kanna et al. (2018).

## Author contribution

YO analyzed data, performed simulations, and produced figures. YO and SA prepared the manuscript. YM developed the model code. SS, NK, DS, and YO contributed to the field work. All authors discussed the results and commented on the manuscript.

## Competing interests

The authors declare that they have no conflict of interest.

## Acknowledgements

We would like to thank Drs. Y. Fukamachi, T. Sawagaki, J. Saito, I. Asaji, and the members of the 2014 and 2016 field campaigns. Special thanks are extended to S. Daorana, T. Ohshima, and K. Peterson for providing logistical support. This research was funded by the Japanese Ministry of Education, Culture, Sports, Science and Technology through the Green Network of Excellence (GRENE) Arctic Climate Change Research Project and the Arctic Challenge for Sustainability (ArCS) Project, and JSPS KAKENHI Grant Number 16K12575.

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



**Table 1.** List of model runs. Run names, initial stratifications, and values of inflow velocity ($V_{sg}$: m s$^{-1}$) and flux of subglacial discharge ($Q_{sg}$: m$^3$ s$^{-1}$).

| Run name | Initial stratification | $V_{sg}$ (m s$^{-1}$) | $Q_{sg}$ (m$^3$ s$^{-1}$) |
|:---:|:---:|:---:|:---:|
| Q100 | Observed in 2014 | 0.01 | 100 |
| CTRL | " | 0.05 | 500 |
| Q600 | " | 0.06 | 600 |
| Q1000 | " | 0.1 | 1000 |
| ST16 | Observed in 2016 | 0.05 | 500 |




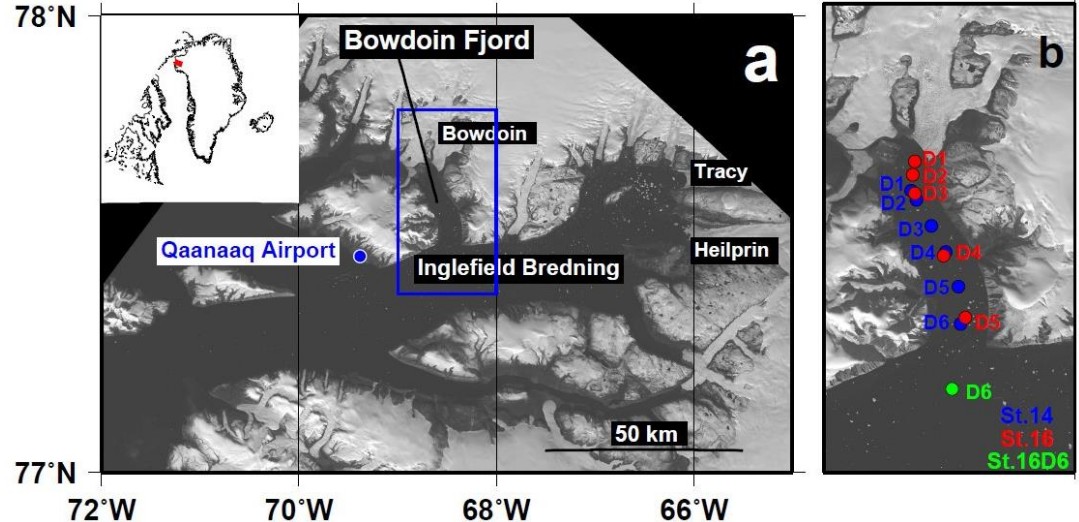

**Figure 1.** Study region in Bowdoin Fjord in northwestern Greenland. (a) Landsat image (6 September 2014) showing northwestern Greenland. The blue box indicates the area shown in (b). The inset shows the location of the region in Greenland. (b) Location of the CTD observation sites indicated by dots (blue in 2014, red in Bowdoin Fjord in 2016, and green located outside Bowdoin Fjord in 2016).




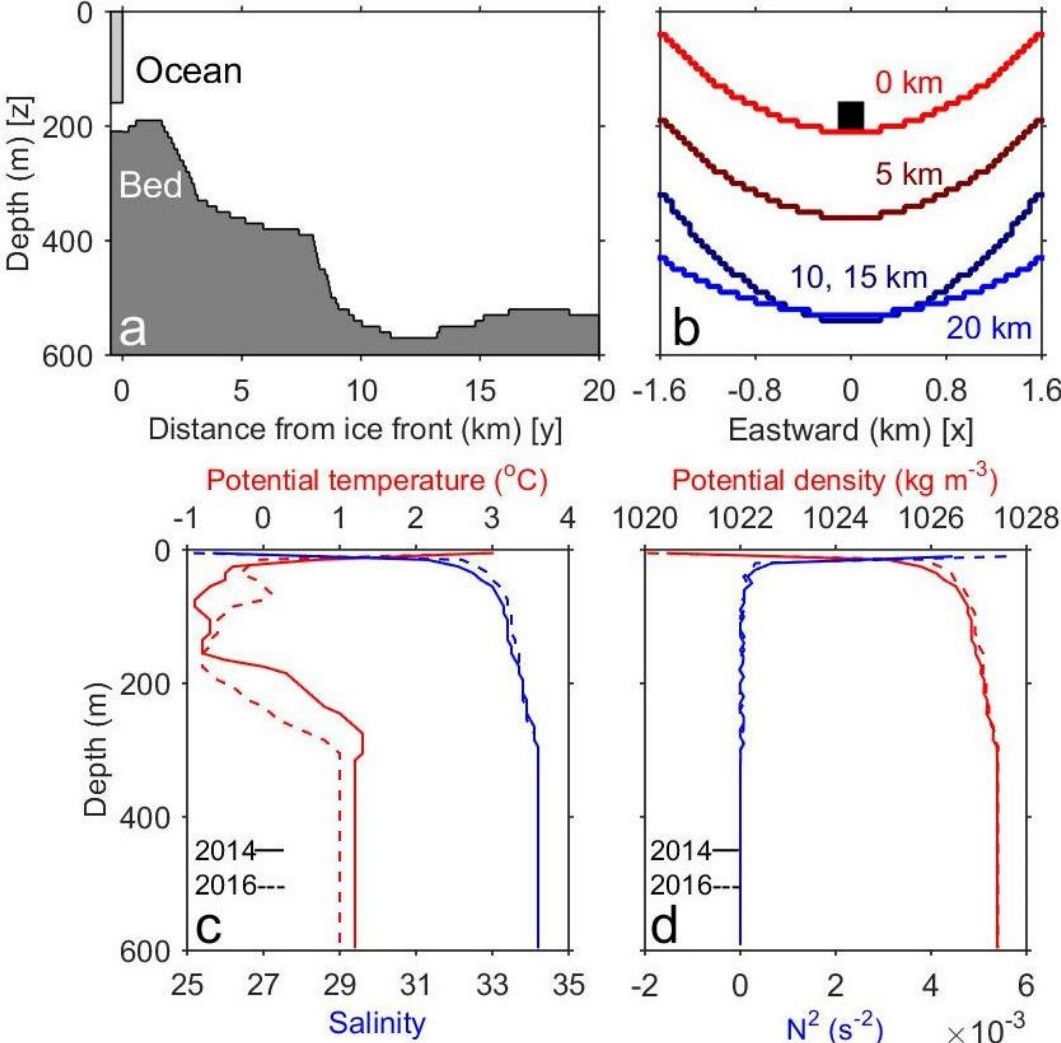

**Figure 2.** Numerical model settings. (a) Ocean depth along the centerline of Bowdoin Fjord (from north to south). The space between the glacier and the sea bed indicates a 50-m-high subglacial drainage conduit. (b) Depth across the fjord (from west to east) at 0, 5, 10, 15, and 20 km from the ice front. The box indicates the subglacial drainage conduit at the center of the fjord (200 m wide × 50 m high; 10000 $m^2$). Initial vertical profiles of (c) potential temperature, salinity, (d) potential density, and the square of Brunt-Väisälä frequency ($N^2$) computed from (c).



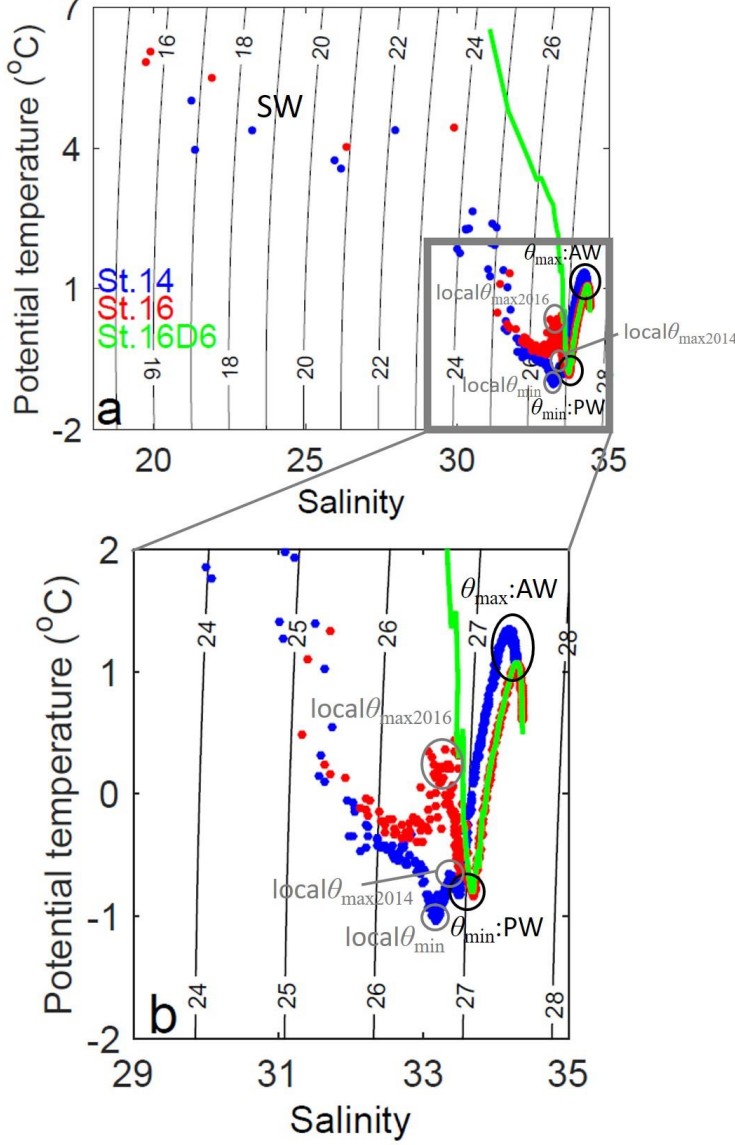

**Figure 3.** (a) Potential temperature–salinity diagrams for the data from 2014 and 2016. Dots are shown at 5 m intervals. The color of the markers corresponds to the sampling sites as indicated in Fig. 1b (blue in 2014, red at Stations 16D1−5, and green at Station 16D6). The potential densities are shown by the black isopycnal contours. The gray box indicates the domain shown in Fig. 8. (b) shows the enlarged region indicated by the gray box in (a).





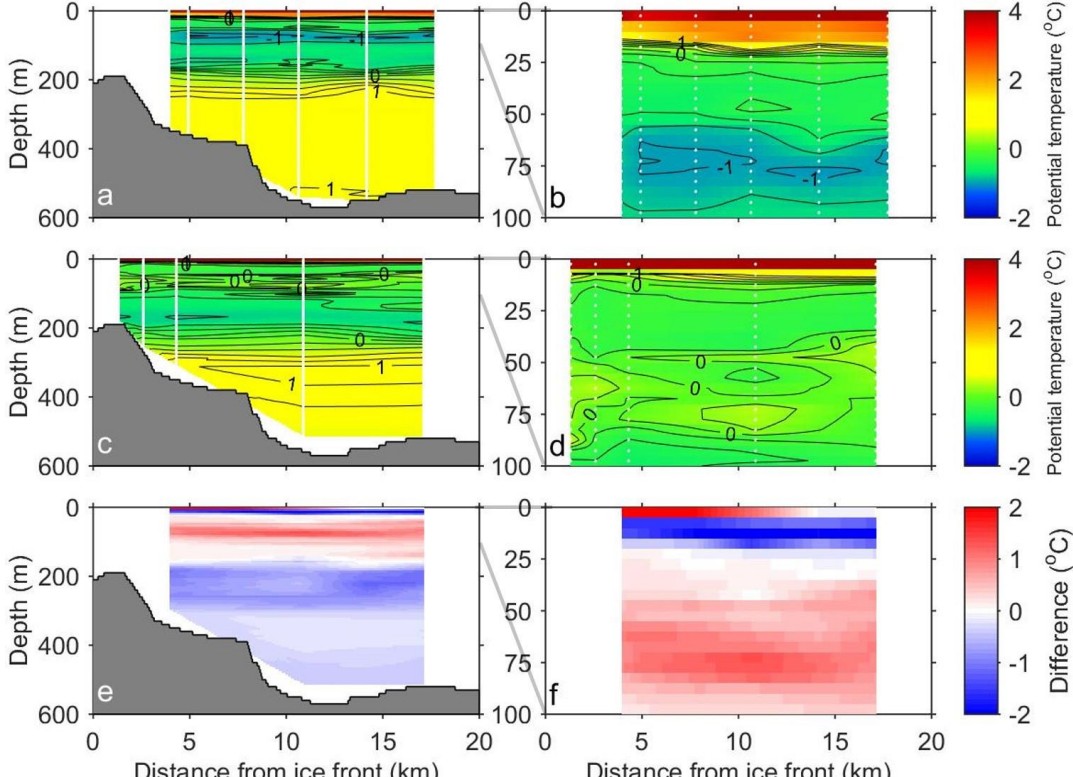

**Figure 4.** Vertical profiles of potential temperature along the centerline of Bowdoin Fjord as observed in (a) 2014 and (c) 2016. The difference between the two years (2016 − 2014) is shown in (e). (b), (d), and (f) show the enlarged region from the sea surface to 100 m.



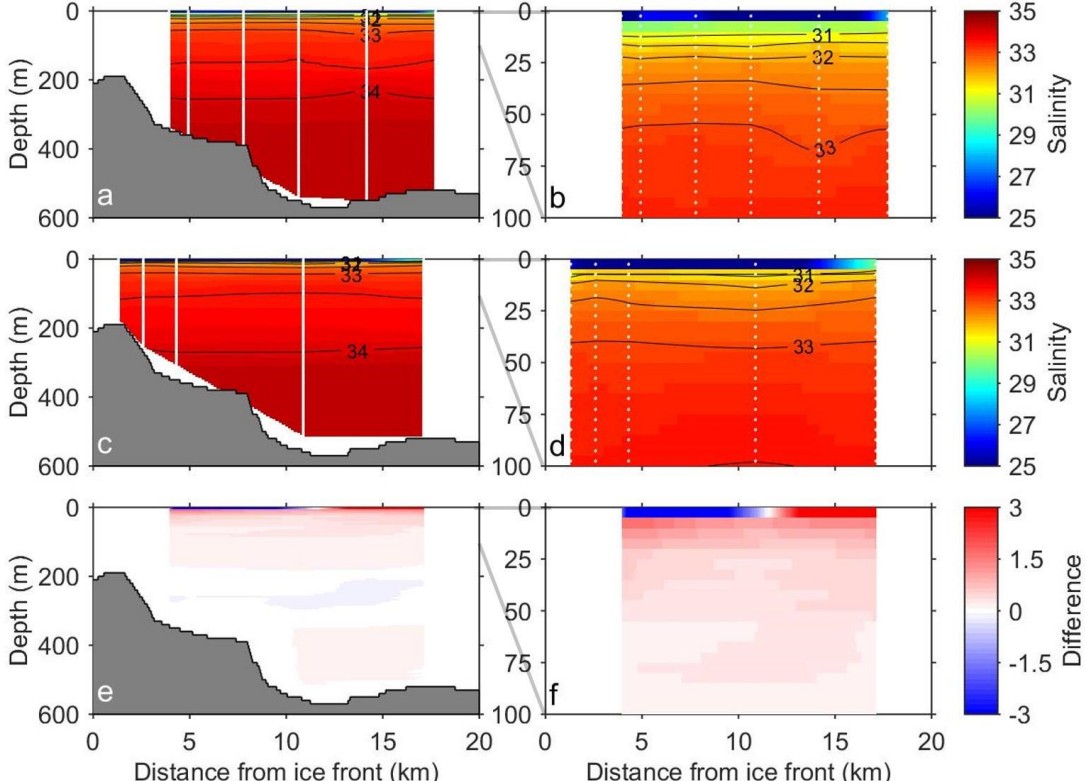

**Figure 5.** Same as Fig. 4, but for salinity.



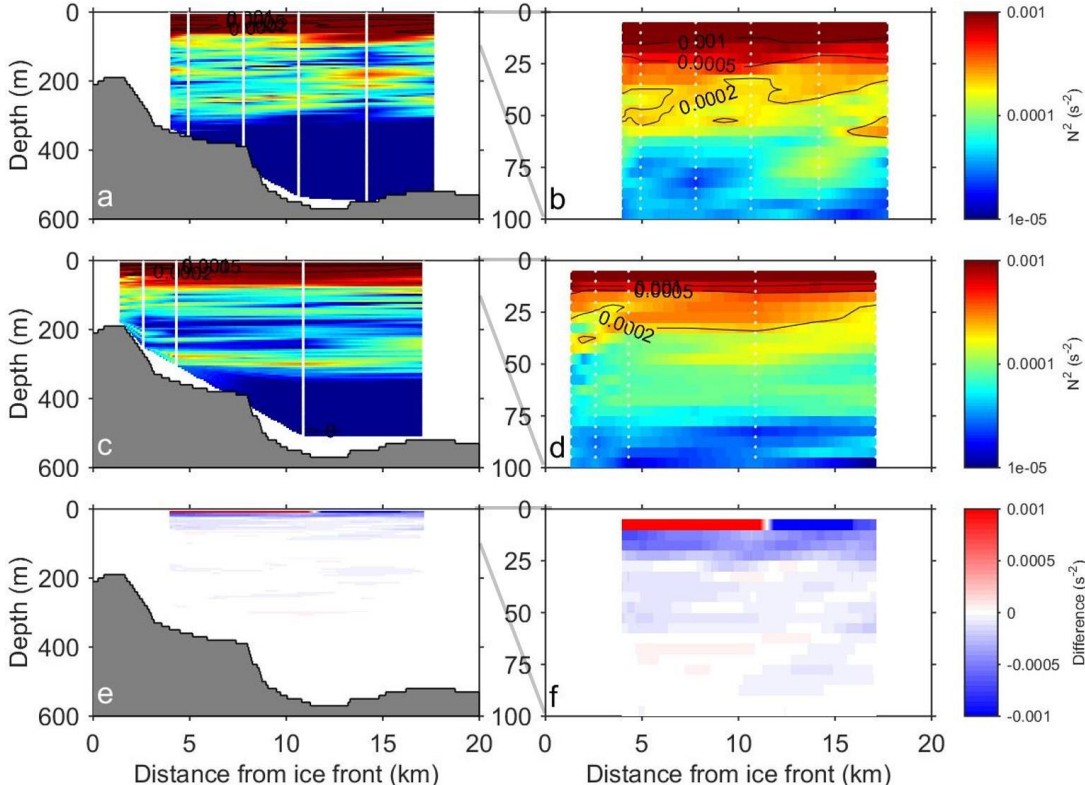

**Figure 6.** Same as Fig. 4, but for square of Brunt-Väisälä frequency ($N^2$).



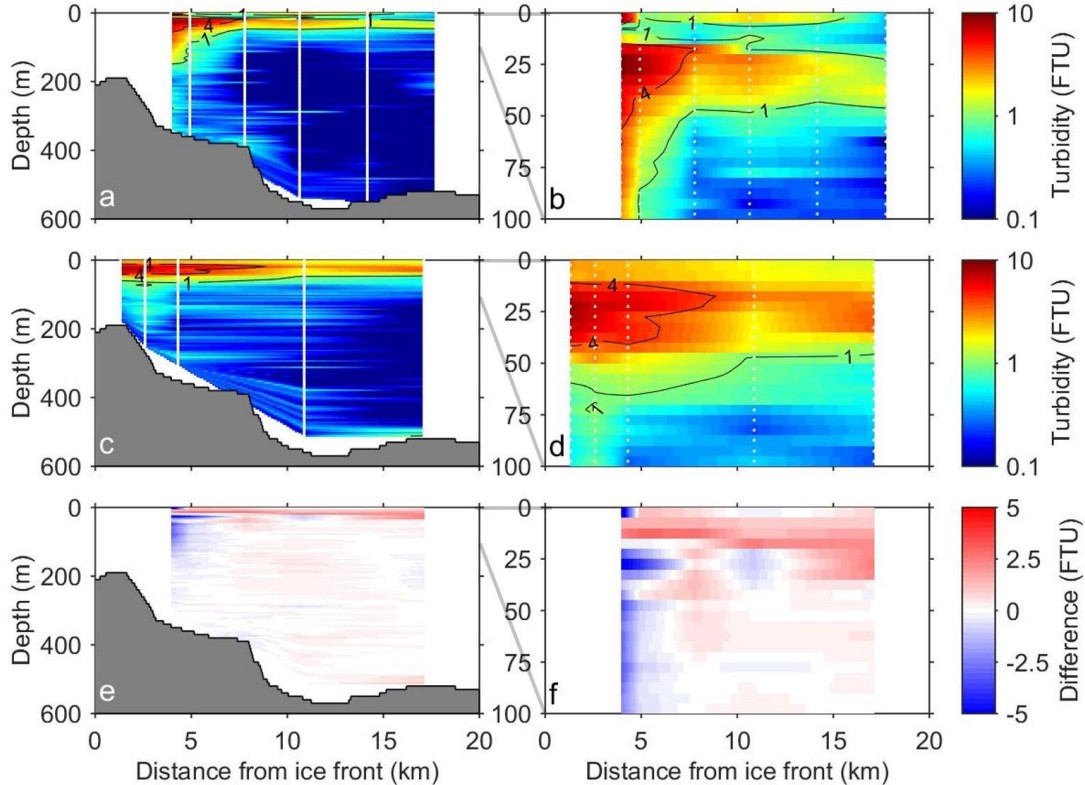

**Figure 7.** Same as Fig. 4, but for turbidity.




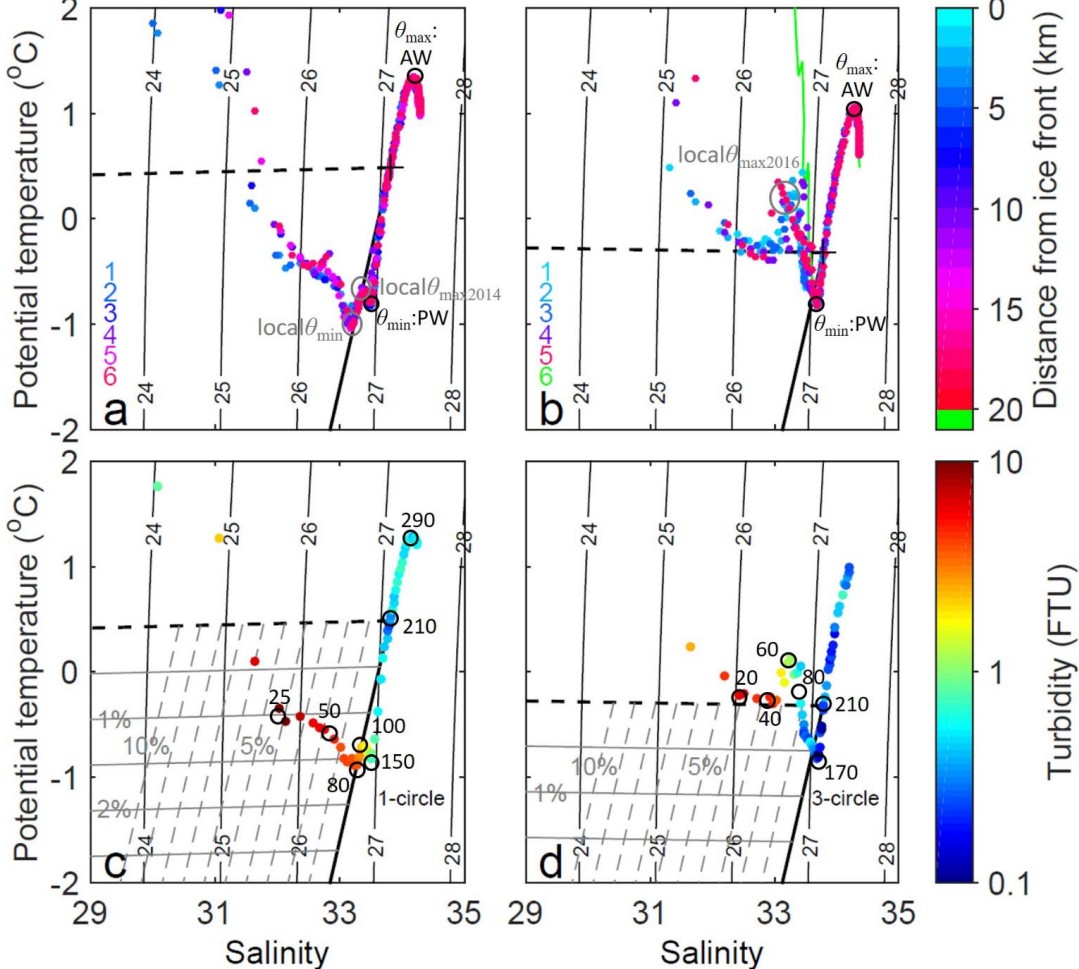

**Figure 8.** Potential temperature–salinity diagrams of the freshwater endmember fractions in (a, c) 2014 and (b, d) 2016 (as
shown in Fig. 3b). The solid and dashed black lines represent the theoretical melt and runoff line, respectively. In (a) and (b),
the data are plotted for stations 1–6, and the color scale indicates the station number. In (c) and (d), the data are plotted for
stations with ~4 km distance from the ice front (14D1 and 16D3 at depths ≥ 5 m). The color of markers denotes turbidity. The
solid and dashed gray lines represent the fractions of submarine meltwater (where the line intervals are 0.5–2.5 %) and
subglacial discharge (where the line intervals are 1–10 %), respectively. The black numbers outside the circles indicate the
depth.

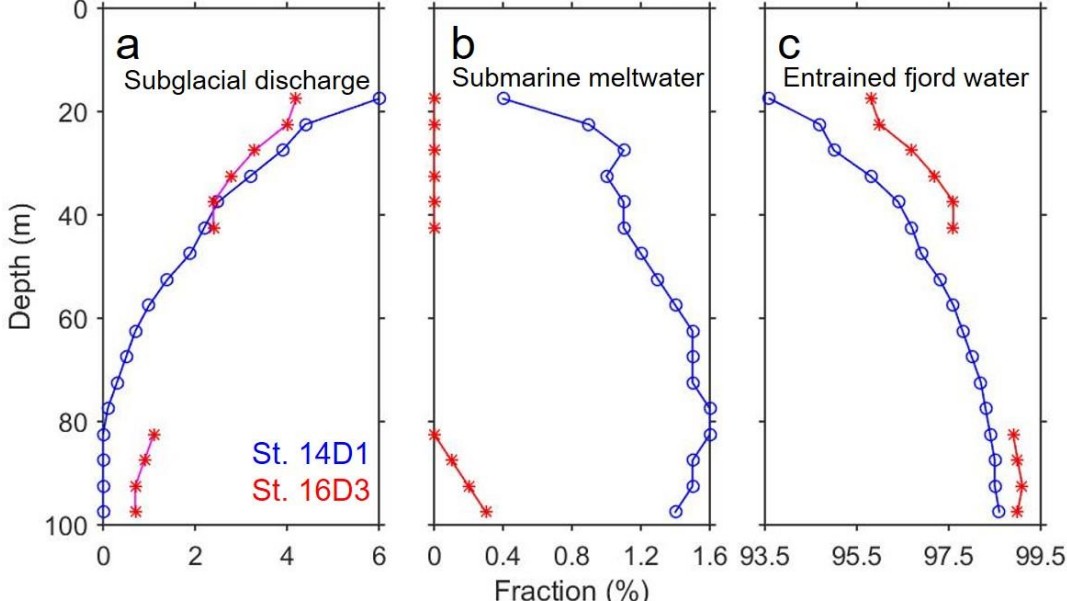

**Figure 9.** Vertical profiles of the three endmember fractions of (a) subglacial discharge, (b) submarine meltwater, and (c) entrained fjord water at Stations 14D1 (blue) and 16D3 (red).




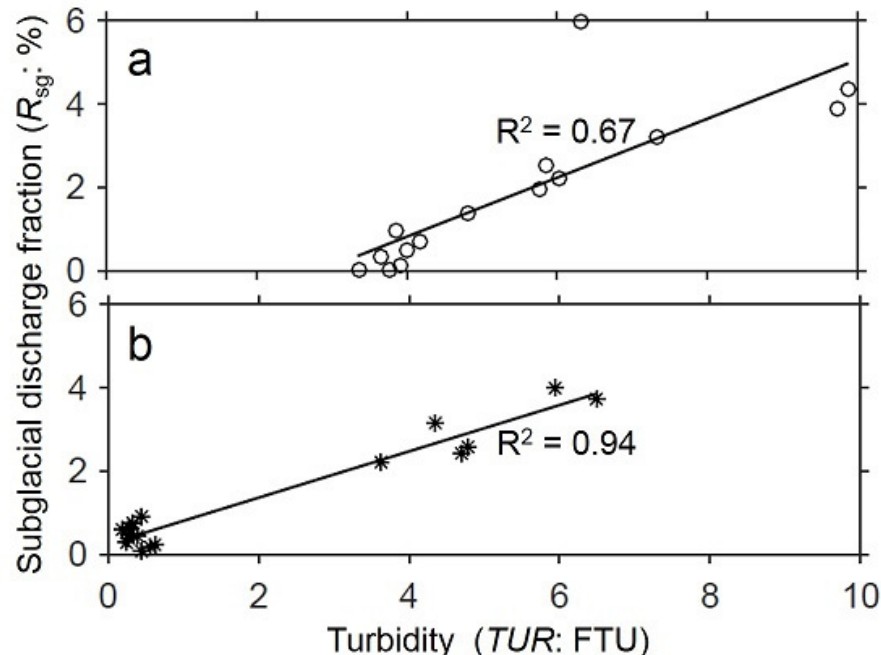

**Figure 10.** Scatter plots of turbidity and the subglacial discharge fraction at Stations (a) 14D1 (data in the meltwater quadrant of Fig. 8c) and (b) 16D3 (data in the meltwater quadrant and that including the data points at a depth of 15–40 m in Fig. 8d). The solid lines represent the linear regression of the data.





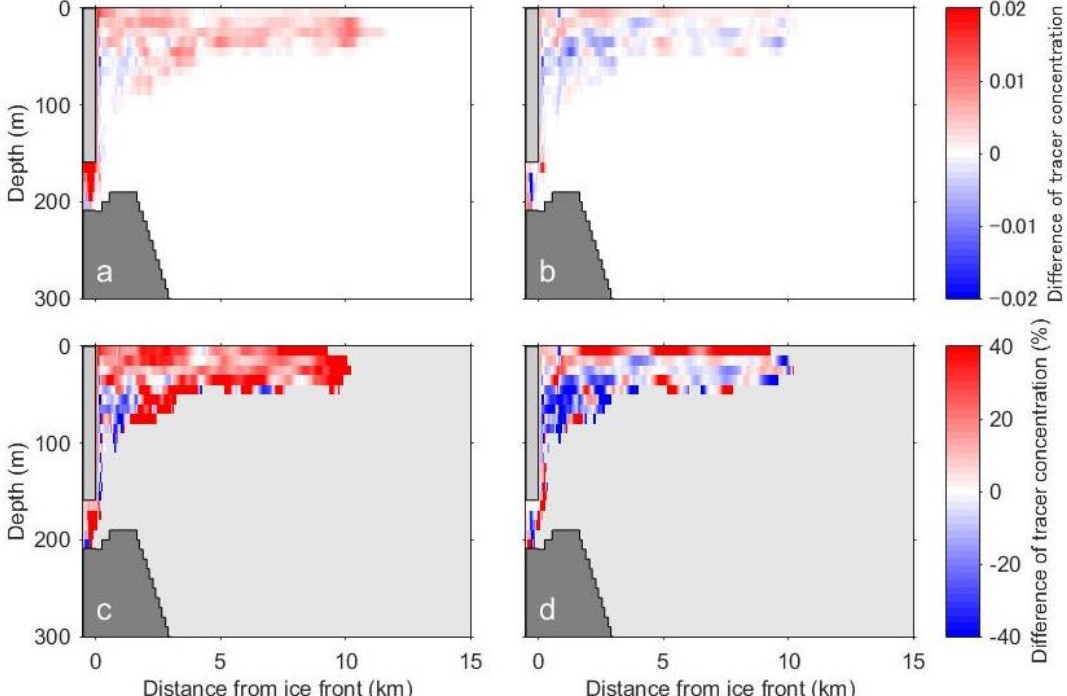

**Figure 11.** (a) Difference between subglacial discharge tracer concentrations for Q600 ($Q_{sg} = 600$ m$^3$ s$^{-1}$) and CTRL ($Q_{sg} = 500$ m$^3$ s$^{-1}$) using the same initial stratification conditions. The difference (Q600 − CTRL) after integration for 15 h is shown for the section along the centerline of the fjord. (b) Difference between the results for the same amount of discharge, but using initial stratifications as observed in 2016 and 2014 (ST16 − CTRL). (c) and (d) Percentage of the results in (a) and (b), respectively.



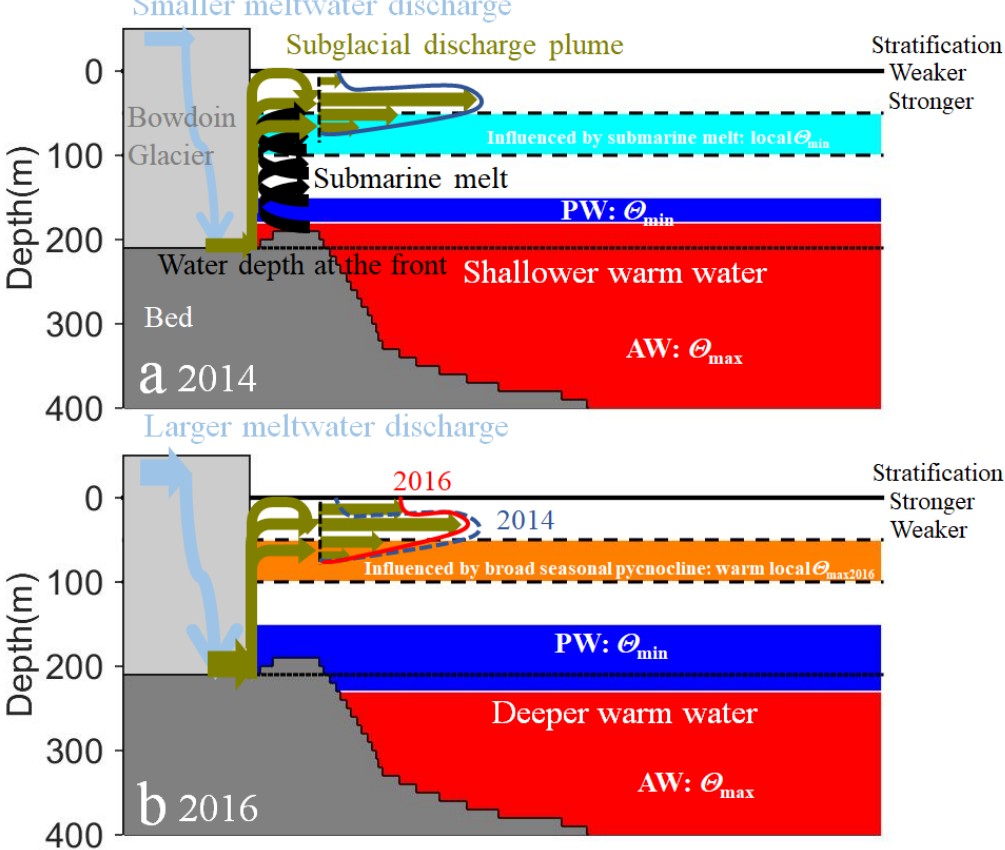

**Figure 12.** Schematic diagram of the possible glacial discharge and water mass structure of Bowdoin Fjord in (a) 2014 and
(b) 2016.




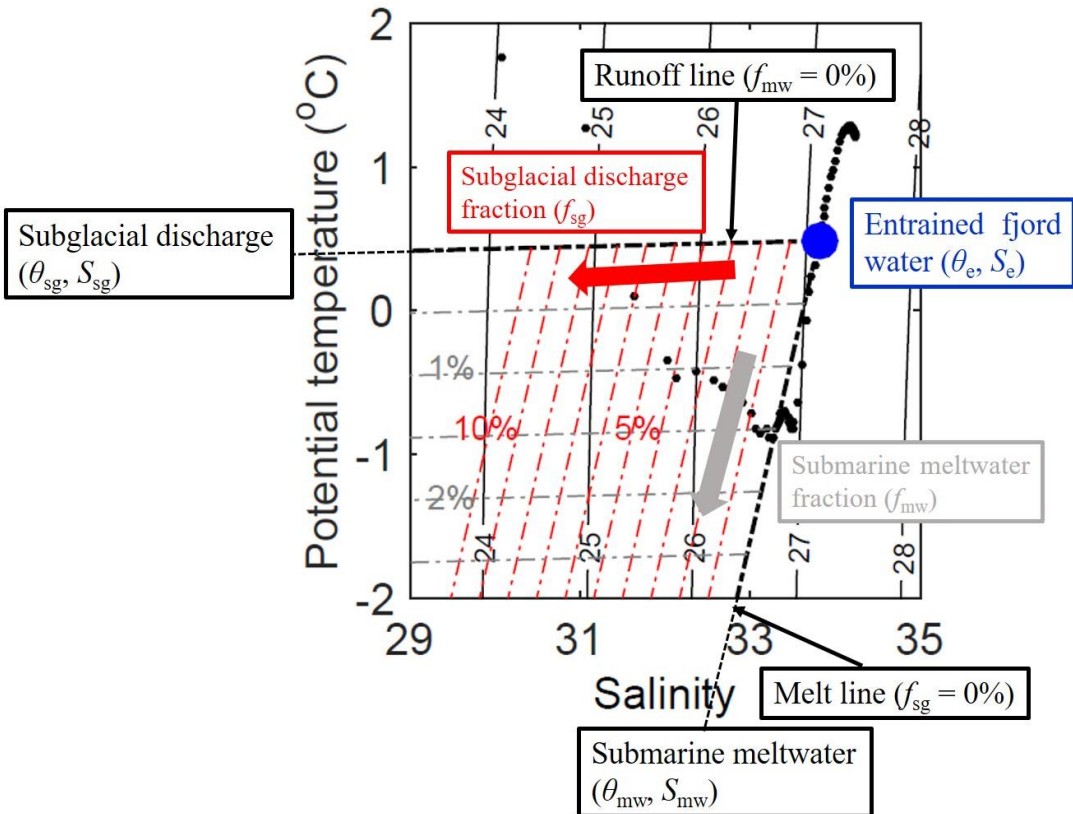

**Figure A1.** Freshwater endmember analysis in the potential temperature–salinity diagram. The black dots represent the data observed at depths ≥ 5 m of Station 14D1.





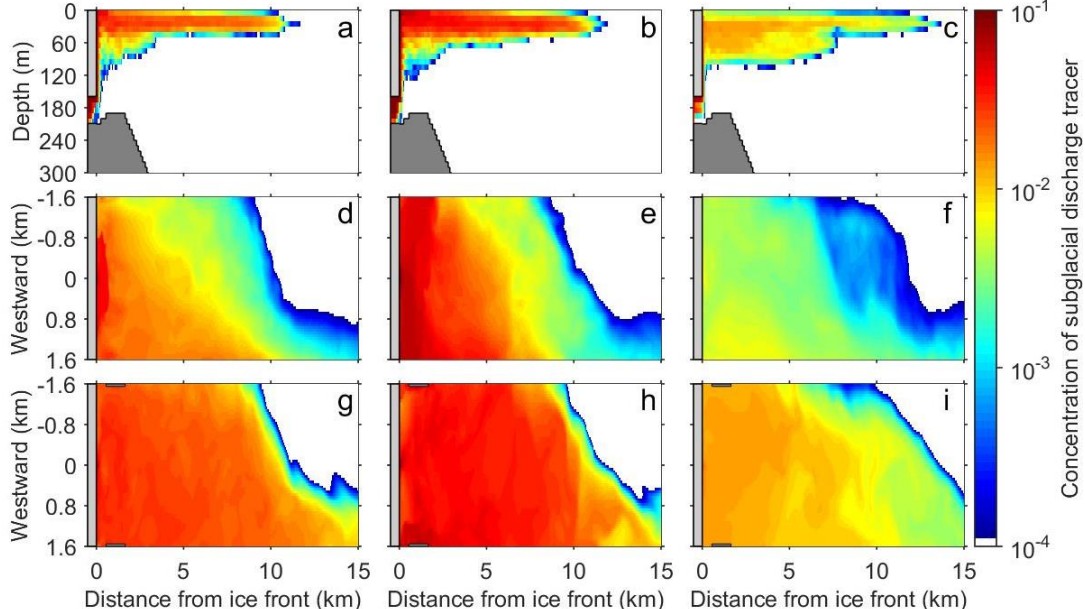

**Figure B1.** (a–c) Vertical profiles of the subglacial discharge tracer concentration along the centerline of the fjord. Horizontal distribution of the tracer concentration (d–f) at the fjord surface (depth: 0–10 m) and (g–i) at a depth of 20–30 m. Experimental results are obtained (a, d, and g) in CTRL after integration for 16 h, (b, e, and h) in Q1000 after 13 h, and (c, f, and i) in Q100 after 56 h. The integration time in each experiment is equivalent to the time required for the first arrival of the tracer.



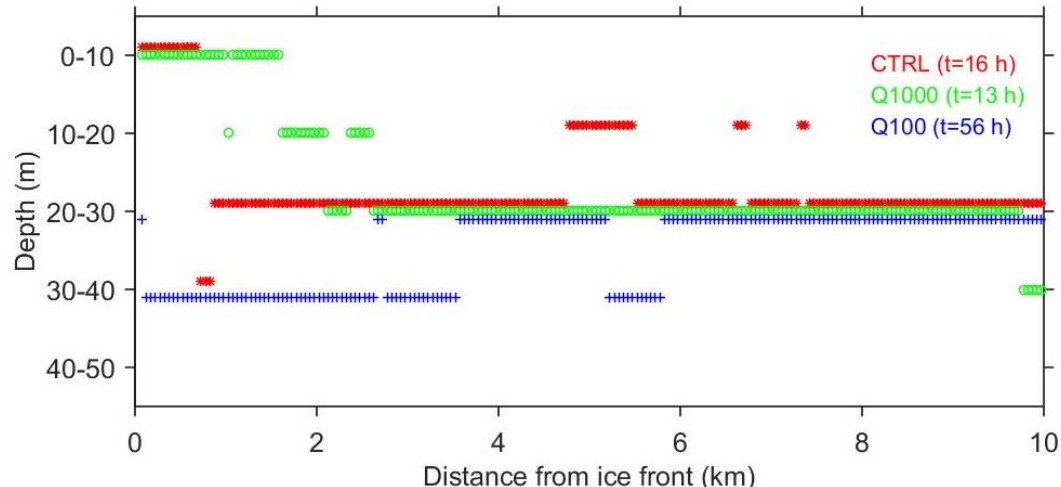

**Figure B2.** Depth of the maximum concentration of subglacial discharge tracer along the centerline of the fjord.