# Peer review of "Vertical distribution of water mass properties under the influence of subglacial discharge in Bowdoin Fjord, northwestern Greenland"

_Ocean Science, 2019_

## Referee Comment (RC1) · Anonymous Referee #1 · 31 May 2019

The manuscript by Ohashi et al. examines water mass structure near a subglacial discharge plume in Bowdoin Fjord. They focus on two sample times (from 2014 and 2016) and compare and contrast the differences between the two samples. They focus on turbidity, in addition to the common/traditional temperature and salinity property analyses. They also run a model to better understand and explain the subglacial discharge and water properties that lead to the observations they have.

I am not a modeler and so I cannot address that part of the paper very well, but overall I think this is a well written paper and the topic and presentation is appropriate to this journal. I have a few major comments (conclusions about sediment visibility at the

surface; code) and some minor comments.

You use Equation 1 and cite Chauché et al. (2014) among others. I think this citation should be removed, because Chauché et al. (2014) does not use the equation that you use. Theirs is different because it contains a typo (a sign change).

Page 3 Line 35/36: Temperature and salinity do not follow the same advection diffusion equation, but you state that they do. Also, should theta here be something other than 0, assuming the subglacial discharge water is at the same temperature as the basal ice, which is not 0 due to pressure dependence of the phase change temperature?

Page 7 Line 42: Seems like you should cite Chu (2012) here when you cite yourself. It seems odd that your 2016 paper also did not cite Chu 2012.

Page 8 line 25: Your 2016 paper used a sensor that was not high enough resolution to distinguish between subglacial discharge and surface streams that deliver sediment to the fjord surface. I disagree with your claim that subglacial and surface stream delivered sediment behave the same. Large(r) volumes of subglacial discharge are more likely to reach the surface and cause the v- or u-shaped sediment plume often seen at the front of marine terminating glaciers. Even larger plumes may make that sediment surface pool even bigger (as you state and support with citations further down the same page). But those pools always disappear rather quickly, and I don't think that even larger subglacial discharge would impact the visual sediments at the fjord surface over large areas as you state in the paragraph beginning on line 25.

Why don't you force your model with RCM runoff estimates? Stevens et al (2016) and Mankoff et al (2016) have shown that RCM runoff estimates do approximately align with estimated runoff derived from plume & fjord observations.

Code Availability: I commend you for releasing some of your code, but I think it is far from sufficient for me to suggest publication of this paper. I downloaded and browsed the Git repository and found it lacking. I'd like to see the following code supplied with

this paper:

1) Code archived at a permanent location such as Zenodo.

2) Zip file extract or GIT tag or something pointing to the specific code that was run, not just a living archive which in a few years may not reflect the code used in this study.

3) README files that are in English that introduce the code, the repository, how to get the specific code version used in this paper, and how to run and reproduce the simulations in this paper. When I try to compile (with pdflatex) the doc/kinaco_doc.tex file, I get a PDF but it really doesn't look correct.

3.1) Documentation of the routines. There are almost no comments in your code!

4) Not just "the model code" but an archive of the configuration and setup that you used to do the work in this paper. Where are the input files? The initial and boundary condition setup? etc. There should be data files and preprocessing scripts included.

5) Non-model code should also be provided. Please package and release your analysis scripts, the code to generate your figures, etc.

Figures: Please read Thyng et al (2016; http://dx.doi.org/10.5670/oceanog.2016.66) and consider re-doing your figures with better colorbars than what appears to be Matlab Jet.

---

## Referee Comment (RC2) · Anonymous Referee #2 · 7 Jul 2019

General Comments:

The manuscript by Ohashi et al. 2019 presents observations from Bowdoin Fjord in northwest Greenland from 2014 and 2016 and supports these observations with interpretations from a 3D nonhydrostatic model. The study is mostly a straightforward report of the measurements between these two years in terms of the stratification and turbidity, which seems to be competent. The observational portion of the work is clear with a topic and discussion that is appropriate for this journal. A scientifically-interesting part is the comparison between the two summers, from which the authors draw conclusions about the influence of variations in the subglacial discharge and stratification. However,

the validity of such a comparison must be addressed with greater care (a few suggestions for this are provided in the specific comments). For instance, might we observe similar differences in turbidity if we simply made the measurements in two different weeks within the same melt season (due to intrinsic variability in the discharge rate)? This also goes hand-in-hand with how the short 5-day integration times in the numerical experiments may be enough to stabilize plumes, but not necessarily enough to set up a steady-state fjord stratification (discussed in specific comments). In addition to integration time, there are also additional issues/caveats that must be addressed with respect to the model configuration (resolution and boundary conditions) use to interpret the observational results. The connection between the model and the observations is also rather tenuous - a direct comparison is not well-established, leaving the authors to draw inferences from a few model experiments to help explain the differences they observe between 2014 and 2016. It is unclear whether the model experiments actually yield any new understanding (of the roles of discharge and stratification) not available from previous studies. If not, it's not clear why the model is needed at all; if so, then why aren't these findings reported as part of the manuscript's results section? Finally, in general, the manuscript is not particularly well written (some technical comments are provided but is not exhaustive) - it could benefit from editing by a native English speaker to improve clarity.

Specific Comments:

Page 1, Line 16: What is the significance of the 60-80 m depth range? What does "temperature profiles were distinctively different" mean specifically? Be more specific about what "a larger fraction" means. In general, this result seems contradictory/unclear: there is more discharge in 2016, but higher discharge fraction at 15-40m in 2014, yet there is also higher turbidity near the surface in 2016. The authors attribute all this to different stratification/discharge flux combinations but it's not at all clear why this is the case from the abstract.

Page 1, Line 29-32 and other lines which use similar citations: This is a subtle point

in the framing of the paper in an atmospheric/ice sheet perspective vs. a growing body of literature on the ocean-driven variability of the ice sheet. While these lines provide an accurate summary of some recent literature on Greenland mass budgets (but also see King et al., 2018: https://doi.org/10.5194/tc-12-3813-2018, Mankoff et al., 2019: https://doi.org/10.5194/essd-11-769-2019), perhaps rewording or additional clarification is necessary to leave the reader with the right impression of the role of ocean in this process. The mass budgets cited use a "flux gate" further up-glacier where surface speed and ice thickness is used to estimate solid ice passing through this gate which is then assumed to be eventually calved into the ocean. This however does not account for surface thinning downstream or submarine melt at the terminus. A short discussion on the ocean's role in undercutting glaciers (through submarine melt) is presented in Rignot et al., 2015: https://doi.org/10.1002/2015GL064236 and Straneo et al., 2015: https://doi.org/10.1146/annurev-marine-010213-135133. This perspective is more relevant for this study than many of the atmosphere/ice references that are presently cited and a discussion focusing on the ocean's role would help set the scene for this study.

Page 1, Line 37-41: While this is an accurate account of recent observations on exiting plume water masses, it should be noted that plume meltwater fractions (cited as 7-10 percent) depends strongly on the discharge strength and depth of the plume source (which together prescribe the degree of entrainment and neutral buoyancy of the exiting plume water mass). Bendtsen and Mankoff's measurements both focus on fjords that are shallow and have a plume undergoing weak overall entrainment i.e. they exit at the surface instead of at mid-depth. In general, the meltwater fractions should be much lower for deeper plume sources or plumes that undergo greater entrainment. Here, you are focusing on a shallow discharge plume that rises to the surface, but it is important to point out that many plumes do not fall into this category and why, as well as why shallow plumes have such a high plume meltwater fraction (see Straneo et al. 2015 for a relevant discussion on plumes: https://doi.org/10.1146/annurev-marine-010213-135133).

Page 2, Line 8: "The subglacial discharge distribution into a fjord" is better described as "vertical distribution of outflowing plume water" since the subglacial discharge only exists as the terminus depth and is not "distributed," but rather the plume outflow can be vertically distributed. There are a few other cases throughout the paper where this distinction could be made clearer. Also, note that the plume outflow distribution cited is more relevant for plumes that reach neutral buoyancy at mid-depths (instead of a concentrated outflow at the surface, as in this study). Consider rewording other instances of this including Page 3, Line 8, and others.

Page 2, Line 11: What is the "realistic influence?" In what way were previous evaluations of its influence not realistic?

Page 5, Line 9: Considering renaming the title to Section 4 as "Observational Results" or similar, since it can be confusing that Sect. 3.3 discusses the numerical experiments, which is not discussed again until Sect. 5.2. An alternative, which may be preferable is restructuring the sections so that all the observational discussion precedes the numerical simulations e.g. Sect. 3: Observational Data and Methods, Sect. 4: Observational Results, Sect. 5: Numerical Experiments.

Page 4, Sect. 3.3 general comments: In general, the numerical experiments would benefit from higher resolution (which would also improve the quality of the numerical results and many of the figures including Fig. 11, B1, and B2) as well as caveats/justifications for certain choices such as the seafloor no-slip conditions and the integration time.

(a) Resolution Issue: There is a strong concern that the results here are strongly dependent on the horizontal and vertical resolution. For instance, the plume (assuming a point plume with entrainment coefficient of alpha = .1, although a similar case can be made for a line plume of finite width) radius at the surface for such cases would be approx. 20m, which is already subgridscale. Many current numerical studies including plumes implement some version of a subgridscale plume parameterization (Xu et al.
2013, Cowton et al. 2015) unless they are extremely high resolution (< 1 m horizontal). Also, not discussed here is why the plume source (choice of outflow dimensions) is chosen to be 200 m wide and 50 m tall. Is there an observational justification for these dimensions or was it chosen such that the source was multiple grid points high and wide? A source that is 50 m tall seems especially large given that the depth of the plume is 210m and much of the change in density of the plume occurs near the source (i.e. within the first 50m). Therefore, it is strongly recommended to run these runs at higher resolution (and to reduce the height of the plume source) to observe how much the results would vary, or demonstrate the convergence of the relevant metrics with respect to resolution.

(b) No-slip conditions: Since the access of the AW is controlled by a small layer (only a few gridpoints tall at 1-2km from the icefront), this choice in bottom boundary condition is likely to dampen the ability of AW to fuel the entrainment from the plume. The choice of no-slip boundary conditions would only be justified if the viscous sublayer were resolved. Consider quadratic drag (which is more defensible) bottom layer or demonstrate insensitivity to/justify the choice of no-slip conditions. What is prescribed at horizontal boundaries? Perhaps an increase in vertical resolution in (a) would help as well.

(c) Integration time: Please consider justifying the integration time of five days (or stating the implications or caveats associated with such a short integration time) or running the numerical simulations for longer. Although the plume rise time is much less than five days, the residence time within the fjord (timescale associated with volume of the fjord divided by rate of overturning circulation) is likely much longer and the response of the background stratification to the water mass transformation due to plume entrainment should be on the timescale of months (see Carroll et al., 2015).

Page 7, Line 42: A caveat which may be worth noting is that turbidity at the fjord surface may only be a reasonable proxy if the plume is able to reach the surface, which depends on the degree of entrainment.

Page 8 (throughout): Consider an additional run ST16Q600, which uses the stratification observed in 2016, with a discharge that is 20 percent higher than the CTRL run, since you state the PDD is 20 percent higher in 2016 compared to 2014, so a more realistic representation of the 2016 state would take into account the increase in discharge as well.

Page 8 (sect. 5.2) : Why are the model results presented in the paper's discussion section - aren't these also results?

Page 22 (also, see other relevant figures and discussions): Each of these measurements represents a snapshot of the turbidity, temp, salinity etc., but are these snapshots representative of the entire 2014 and 2016 melt seasons? How much variability in the discharge rate, turbidity etc. would we expect within a single melt season? I'm not sure whether it's possible to even distinguish differences between the 2014 and 2016 melt seasons from a single sample from each season. If not, then attempting to explain them using the model is not a valid approach.

Technical Corrections:

Page 1, Line 1 and many others. When "structures" is used to denote the properties of vertical water mass profiles i.e. "water structures", this should instead be "vertical density profiles" or equivalent for clarity. For instance, consider changing the title to: "The effect of subglacial discharge on vertical density profiles in Bowdoin Fjord, northwestern Greenland."

Page 1, Line 23: Consider rewording the last sentence of the abstract for clarity e.g.: "Fjord stratification is an important factor controlling the vertical distribution of freshwater outflow due to subglacial discharge strength and entrainment. The fjord stratification does not influence the magnitude of subglacial discharge 'amount', as is implied in the original statement.

Page 2, Line 34: "...of a proglacial fjord." Page 2, Line 42: Correct to present tense:

"This study focuses on BF . . .."

Page 3, Line 1: Correct to: "In June and July, the sea ice melts rapidly exposing the open ocean surface."

Page 3, Line 6. If the depth of PW/AW interface is between 50m and 150m, then 210m should always be below the AW and PW interface. In Fig. 12, the schematic shows a PW/AW interface that is between 175m and 225m.

Page 3, Line 24: "0.01" psu? Page 3, Line 36: ". . .which subsequently spreads due to entrainment."

Page 4, Line 9 and others: Consider another word instead of "endmember" which is unclear when it is used. . . perhaps "source?"

Page 5, Line 27 and others: Consider a more compact notation $\Theta_{\max}^{2014}$, etc.

Page 9, Line 29: As a point of clarification, the subglacial discharge that exits near the surface as in this study would lighten the surface layers and increase stratification, but if it preferentially lightens intermediate/deeper layers, it would act to decrease stratification as is observed in other studies (see Jackson et al., 2017: https://doi.org/10.1002/2017GL073602).

Page 9, Line 43: "In turn, a prominent local . . . was detected around 60 m in 2016, which was significantly warmer than the local maximum in 2014."

Page 16, Line 4: Correct to: ". . . (blue in 2014, red in 2016, both inside Bowdoin Fjord, and green in 2016, outside Bowdoin Fjord)."

Fig.12 and to a lesser extent, Fig. 13: Consider esthetic improvements that would improve this figure including fewers arrows, clearer color/font contrast, labels, etc. Is the PW layer here meant to only represent the PW core? It may be clearer to differentiate between $\Theta_{\min}$ and the full PW layer. It is not clear exactly which portion of the vertical column has a stronger vs. weaker stratification.

---

## Author Comment (AC1) · 9 Dec 2019

Revision of the manuscript [Paper #os-2019-33] "Water mass structure and the effect of subglacial discharge in Bowdoin Fjord, northwestern Greenland" by Ohashi et al.

We thank editor and the two referees for careful reading of our manuscript and for giving useful comments and suggestions. We have addressed the raised referee comments and explained how we revised. The major changes are listed as follows.

・Title:

 "Vertical distribution of water mass properties under the influence of subglacial discharge in Bowdoin Fjord, northwestern Greenland"

・Structure of the paper:

Sect. 1: Introduction

Sect. 2: Study area

Sect. 3: Observational data and methods

Sect. 4: Observational results

Sect. 5: Numerical experiments

5.1 Experimental settings

5.2 Experimental results

Sect. 6: Discussion

Sect. 7: Conclusions

The referee comments are written in bold and replies from the authors follow the referee comments. Changes are highlighted in the manuscript using the blue color (response to comment from Referee #1) and green color (Referee #2).
* * *
Response to Comment from Referee #1

**Anonymous Referee #1**

**The manuscript by Ohashi et al. examines water mass structure near a subglacial discharge plume in Bowdoin Fjord. They focus on two sample times (from 2014 and 2016) and compare and contrast the differences between the two samples. They focus on turbidity, in addition to the common/traditional temperature and salinity property analyses. They also run a model to better understand and explain the subglacial discharge and water properties that lead to the observations they have. I am not a modeler and so I cannot address that part of the paper very well, but overall I think this is a well written paper and the topic and presentation is appropriate to this journal. I have a few major comments (conclusions about sediment visibility at the surface; code) and some minor comments.**

Thank you very much for your positive assessment and constructive comments. We have addressed

the main points as follows and responded to all comments.

- We have recalculated the fraction and reperformed the the numerical experiments using the pressure dependent temperature as the temperature of subglacial discharge.
- We have reperformed the numerical experiments using the RACMO output.
- We have uploaded the model source code including documentation of the routines and non-model codes to Zenodo.

**You use Equation 1 and cite Chauché et al. (2014) among others. I think this citation should be removed, because Chauché et al. (2014) does not use the equation that you use. Theirs is different because it contains a typo (a sign change).**

We have removed Chauché et al. (2014) in Page 4 Line 12.

**Page 3 Line 35/36: Temperature and salinity do not follow the same advection diffusion equation, but you state that they do. Also, should theta here be something other than 0, assuming the subglacial discharge water is at the same temperature as the basal ice, which is not 0 due to pressure dependence of the phase change temperature?**

Since the grid spacing of the present model does not resolve molecular diffusivity, the diffusivity referred in the text is actually the eddy diffusivity, whose origin is the net transport by subgrid-scale turbulent eddies. Therefore, the diffusivity coefficients for both temperature and salinity should be the same value in the present model.

Also, we have recalculated the freshwater fraction and reperformed the numerical experiments using $\theta_{sg} = -0.15\ °C$ as the temperature of subglacial discharge due to the pressure dependence of the phase change temperature. Please see Page 4 Lines 6–7 and Page 7 Line 22.

**Page 7 Line 42: Seems like you should cite Chu (2012) here when you cite yourself. It seems odd that your 2016 paper also did not cite Chu 2012.**

According to your comment, we have cited Chu et al. (2012) in Page 8 Line 32. Chu et al. (2012) focused on the margin of confined proglacial river and narrow fjord, and its environment is therefore significantly different from that of the open ocean in Ohashi et al. (2016). Thus, Ohashi et al. (2016) did not cite Chu et al. (2012). In this paper, we think that Chu et al. (2012) is more proper citation than Ohashi et al. (2016).

**Page 8 line 25: Your 2016 paper used a sensor that was not high enough resolution to distinguish between subglacial discharge and surface streams that deliver sediment to the fjord surface. I disagree with your claim that subglacial and surface stream delivered sediment behave the same. Large(r) volumes of subglacial discharge are more likely to reach the surface and cause the v-**

or u-shaped sediment plume often seen at the front of marine terminating glaciers. Even larger plumes may make that sediment surface pool even bigger (as you state and support with citations further down the same page). But those pools always disappear rather quickly, and I don't think that even larger subglacial discharge would impact the visual sediments at the fjord surface over large areas as you state in the paragraph beginning on line 25.

We agree with your suggestion and have removed this part in Page 9 Lines 21–23.

Why don't you force your model with RCM runoff estimates? Stevens et al (2016) and Mankoff et al (2016) have shown that RCM runoff estimates do approximately align with estimated runoff derived from plume & fjord observations.

According to your suggestion, we have reperformed the numerical experiments using the subglacial discharge flux estimated from the Regional Atmospheric Climate Model (RACMO) runoff data (Fig. R1) (Please see Page 7 Lines 34–36). Although the amount of RACMO discharge was an order smaller than those previously assumed, the qualitative result was basically the same. We have added the part on the subglacial discharge estimates in Page 3 Lines 14–20 as follows.

"To help estimate the subglacial discharge conditions for BG, we used Regional Atmospheric Climate Model (RACMO) 2.3p2 runoff data downscaled to 1 km (Noël et al., 2016; Noël et al., 2018). The catchment of the glacier was determined from the Greenland Ice Mapping Project digital elevation model (Howat et al., 2014). In summer (June–August), the daily mean amount of subglacial discharge estimated from the RACMO data varied greatly ($19 \pm 16$ m$^3$ s$^{-1}$ in 2014 and $21 \pm 18$ m$^3$ s$^{-1}$ in 2016). Using the daily mean values of the five days prior to the observation dates (Mankoff et al., 2016), we perform numerical experiments (see Sect. 5 for details). The daily mean amount of subglacial discharge over five days in 2016 (45 m$^3$ s$^{-1}$) was estimated to be 25 % greater than that in 2014 (36 m$^3$ s$^{-1}$)."

[Figure]

Figure R1. Daily subglacial discharge estimates of Bowdoin Glacier in (a) 2014 and (b) 2016. Gray bar shows the five days prior to the observation date.

**Code Availability: I commend you for releasing some of your code, but I think it is far from sufficient for me to suggest publication of this paper. I downloaded and browsed the Git repository and found it lacking. I'd like to see the following code supplied with this paper:**

1) **Code archived at a permanent location such as Zenodo.**

2) **Zip file extract or GIT tag or something pointing to the specific code that was run, not just a living archive which in a few years may not reflect the code used in this study.**

**3.1) Documentation of the routines. There are almost no comments in your code!**

**4) Not just "the model code" but an archive of the configuration and setup that you used to do the work in this paper. Where are the input files? The initial and boundary condition setup? etc. There should be data files and preprocessing scripts included**

According to your comments on code availability (1-4), we have added the English version of the documentation of the routines (README_en), which explains how to compile/build/run the model, to the git repository published at http://lmr.aori.u-tokyo.ac.jp/feog/ymatsu/kinaco.git. Additionally, we have uploaded the used version of model source code, the configuration namelist file, all input files, and preprocessing scripts to Zenodo (https://zenodo.org/record/3532803). Please see Page 12 "Code availability" and "Data availability".

**5)Non-model code should also be provided. Please package and release your analysis scripts, the code to generate your figures, etc.**

We have packaged and uploaded analysis scripts and the codes to generate the figures to Zenodo (https://zenodo.org/record/3532803).

**Figures: Please read Thyng et al (2016; http://dx.doi.org/10.5670/oceanog.2016.66) and consider re-doing your figures with better colorbars than what appears to be Matlab Jet.**

According to Thyng et al. (2016), we have changed the colorbars of figures 3-7, 10, and C1.
* * *
Response to Comment from Referee #2

**Anonymous Referee #2**

**General Comments:**

**The manuscript by Ohashi et al.   2019 presents observations from Bowdoin Fjord in northwest Greenland from 2014 and 2016 and supports these observations with inter- pretations from a 3D nonhydrostatic model. The study is mostly a straightforward report of the measurements between these two years in terms of the stratification and turbid- ity, which seems to be competent. The observational portion of the work is clear with a topic and discussion that is appropriate for this journal. A scientifically-interesting part is the comparison between the two summers, from which the authors draw conclusions about the influence of variations in the**

subglacial discharge and stratification. However, the validity of such a comparison must be addressed with greater care (a few sugges- tions for this are provided in the specific comments). For instance, might we observe similar differences in turbidity if we simply made the measurements in two different weeks within the same melt season (due to intrinsic variability in the discharge rate)? This also goes hand-in-hand with how the short 5-day integration times in the numeri- cal experiments may be enough to stabilize plumes, but not necessarily enough to set up a steady-state fjord stratification (discussed in specific comments). In addition to integration time, there are also additional issues/caveats that must be addressed with respect to the model configuration (resolution and boundary conditions) use to interpret the observational results. The connection between the model and the observations is also rather tenuous - a direct comparison is not well-established, leaving the authors to draw inferences from a few model experiments to help explain the differences they observe between 2014 and 2016. It is unclear whether the model experiments actually yield any new understanding (of the roles of discharge and stratification) not available from previous studies. If not, it's not clear why the model is needed at all; if so, then why aren't these findings reported as part of the manuscript's results section? Finally, in general, the manuscript is not particularly well written (some technical comments are provided but is not exhaustive) - it could benefit from editing by a native English speaker to improve clarity.

We would like to thank you very much for your thorough reading of our manuscript and providing very constructive comments. We have addressed the main points as follows to respond your comment.

・We have reperformed the numerical experiments with changing the settings.

・We have added the part of model results.

**Specific Comments:**

**Page 1, Line 16: What is the significance of the 60-80 m depth range? What does "tem- perature profiles were distinctively different" mean specifically? Be more specific about what "a larger fraction" means. In general, this result seems contradictory/unclear: there is more discharge in 2016, but higher discharge fraction at 15-40m in 2014, yet there is also higher turbidity near the surface in 2016. The authors attribute all this to different stratification/discharge flux combinations but it's not at all clear why this is the case from the abstract.**

This sentence represents the depth that submarine melting layer and seasonal pycnocline layer developed (Please see Sects. 6.3 and 7). However, because this sentence is unclear, we have removed this sentence from abstract.

**Page 1, Line 29-32 and other lines which use similar citations: This is a subtle point in the framing of the paper in an atmospheric/ice sheet perspective vs. a growing body of literature on the ocean-driven variability of the ice sheet. While these lines provide an accurate summary of some recent literature on Greenland mass budgets (but also see King et al., 2018: https://doi.org/10.5194/tc-12-3813-2018, Mankoff et al., 2019: https://doi.org/10.5194/essd-11-769-2019), perhaps rewording or additional clarification is necessary to leave the reader with the right impression of the role of ocean in this process. The mass budgets cited use a "flux gate" further up-glacier where surface speed and ice thickness is used to estimate solid ice passing through this gate which is then assumed to be eventually calved into the ocean. This however does not account for surface thinning downstream or submarine melt at the terminus. A short discussion on the ocean's role in undercutting glaciers (through submarine melt) is presented in Rignot et al., 2015: https://doi.org/10.1002/2015GL064236 and Straneo et al., 2015: https://doi.org/10.1146/annurev-marine-010213-135133. This perspective is more relevant for this study than many of the atmosphere/ice references that are presently cited and a discussion focusing on the ocean's role would help set the scene for this study.**

To clarify the role of ocean on Greenland mass budgets, we have rewrote the introduction in Page 1 Lines 29−33 as follows.

"Increased ice discharge has been potentially induced by increased submarine melting and iceberg calving (Catania et al., 2018; Motyka et al., 2011; Straneo and Heimbach, 2013). At the ice front of marine-terminating outlet glacier, surface melt-induced meltwater discharge drives the rapid submarine melting (e.g., Motyka et al., 2013) and hence promote to iceberg calving due to undercutting of the ice front (O'Leary and Christoffersen, 2013; Rignot et al., 2015). Thus, meltwater discharge from marine-terminating glaciers plays an important role in controlling ice loss."

**Page 1, Line 37-41: While this is an accurate account of recent observations on exiting plume water masses, it should be noted that plume meltwater fractions (cited as 7-10 percent) depends strongly on the discharge strength and depth of the plume source (which together prescribe the degree of entrainment and neutral buoyancy of the exiting plume water mass). Bendtsen and Mankoff's measurements both focus on fjords that are shallow and have a plume undergoing weak overall entrainment i.e. they exit at the surface instead of at mid-depth. In general, the meltwater fractions should be much lower for deeper plume sources or plumes that undergo greater entrainment. Here, you are focusing on a shallow discharge plume that rises to the surface, but it is important to point out that many plumes do not fall into this category and why, as well as why shallow plumes have such a high plume meltwater fraction (see Straneo et al.**

**2015 for a relevant discussion on plumes: https://doi.org/10.1146/annurev-marine-010213-135133).**

We have added the point that why many plumes cannot reach the surface and why shallow plumes have a high plume meltwater fraction in Page 1 Line 38–Page 2 Line 2 as follows.

"Observations in the Greenlandic shallow fjords have revealed that plume surface waters consist of 7–10 % subglacial discharge and ~90 % entrained fjord waters (Bendtsen et al., 2015; Mankoff et al., 2016), indicating that subglacial discharge plumes transport significant amounts of ambient deep water to the fjord surface. The plumes have higher fraction of subglacial discharge than those in the deep fjords due to greater entrainment of ambient water. Additionally, in the deep fjords, many plumes cannot reach the surface because the plumes become neutrally buoyant before reaching the surface (Straneo and Cenedese, 2015)."

**Page 2, Line 8: "The subglacial discharge distribution into a fjord" is better described as "vertical distribution of outflowing plume water" since the subglacial discharge only exists as the terminus depth and is not "distributed," but rather the plume outflow can be vertically distributed. There are a few other cases throughout the paper where this distinction could be made clearer. Also, note that the plume outflow distribution cited is more relevant for plumes that reach neutral buoyancy at mid-depths (instead of a concentrated outflow at the surface, as in this study). Consider rewording other instances of this including Page 3, Line 8, and others.**

We have changed "the subglacial discharge distribution" to "vertical distribution of outflowing plume water" through the manuscript.

**Page 2, Line 11: What is the "realistic influence?" In what way were previous evalua- tions of its influence not realistic?**

The sentence "The realistic influence on the subglacial discharge distribution has not been assessed in detail" was misleading. Chauché et al. (2014) did not perform the numerical experiments and did not calculate the subglacial discharge fraction. Thus, they cannot quantitatively assess the impact of change in the amount of discharge on subglacial discharge fraction change. We mean to say that the relationship between the change in the amount of discharge and observed subglacial discharge fraction has not been assessed quantitatively. We have changed "The realistic influence on the subglacial discharge distribution has not been assessed in detail." to "The relationship with observed subglacial discharge fraction has not been assessed quantitatively." in Page 2 Line 15.

**Page 5, Line 9: Considering renaming the title to Section 4 as "Observational Results" or similar,**

**since it can be confusing that Sect. 3.3 discusses the numerical experiments, which is not discussed again until Sect. 5.2. An alternative, which may be preferable is restructuring the sections so that all the observational discussion precedes the numer- ical simulations e.g. Sect. 3: Observational Data and Methods, Sect. 4: Observational Results, Sect. 5: Numerical Experiments.**

We have added Sect. 5 and changed the structure of paper as follows.

Sect. 3: Observational data and methods

Sect. 4: Observational results

Sect. 5: Numerical experiments

5.1 Experimental settings

5.2 Experimental results

**Page 4, Sect. 3.3 general comments: In general, the numerical experiments would benefit from higher resolution (which would also improve the quality of the numer- ical results and many of the figures including Fig. 11, B1, and B2) as well as caveats/justifications for certain choices such as the seafloor no-slip conditions and the integration time.**

**(a) Resolution Issue: There is a strong concern that the results here are strongly de- pendent on the horizontal and vertical resolution. For instance, the plume (assuming a point plume with entrainment coefficient of alpha = .1, although a similar case can be made for a line plume of finite width) radius at the surface for such cases would be approx. 20m, which is already subgridscale. Many current numerical studies including plumes implement some version of a subgridscale plume parameterization (Xu et al. 2013, Cowton et al. 2015) unless they are extremely high resolution (< 1 m horizontal). Also, not discussed here is why the plume source (choice of outflow dimensions) is chosen to be 200 m wide and 50 m tall. Is there an observational justification for these dimensions or was it chosen such that the source was multiple grid points high and wide? A source that is 50 m tall seems especially large given that the depth of the plume is 210m and much of the change in density of the plume occurs near the source (i.e. within the first 50m). Therefore, it is strongly recommended to run these runs at higher resolution (and to reduce the height of the plume source) to observe how much the results would vary, or demonstrate the convergence of the relevant metrics with respect to resolution.**

According to your comment, we have changed the model resolution (20 m horizontally, 5 m vertically) and reduced the height and width of the plume source (40 m wide × 10 m high; 400 m²). Please see Page 7 Lines 16–18.

**(b) No-slip conditions: Since the access of the AW is controlled by a small layer (only a few grid points tall at 1-2km from the ice front), this choice in bottom boundary condition is likely to dampen the ability of AW to fuel the entrainment from the plume. The choice of no-slip boundary conditions would only be justified if the viscous sublayer were resolved. Consider quadratic drag (which is more defensible) bottom layer or demonstrate insensitivity to/justify the choice of no-slip conditions. What is prescribed at horizontal boundaries? Perhaps an increase in vertical resolution in (a) would help as well.**

We have considered quadratic drag bottom layer. The quadratic drag coefficient is set to $2.5 \times 10^{-3}$. Please see Page 7 Lines 27−28. Also, rigid wall boundary conditions are used for eastern and western ends.

**(c) Integration time: Please consider justifying the integration time of five days (or stating the implications or caveats associated with such a short integration time) or running the numerical simulations for longer. Although the plume rise time is much less than five days, the residence time within the fjord (timescale associated with volume of the fjord divided by rate of overturning circulation) is likely much longer and the response of the background stratification to the water mass transformation due to plume entrainment should be on the timescale of months (see Carroll et al., 2015).**

Because we have no observational data before subglacial discharge, it is difficult to assess the long-term formation process of observed fjord stratification influenced by subglacial discharge in detail. Thus, we have focused on the short-term transitional process of the subglacial discharge in the numerical experiments to interpret the observational data. The model is integrated for seven days from a state of rest, which is sufficient time for the subglacial discharge tracer to reach the southern boundary. We have added the caveats associated with such a short integration time as follows.

In Page 7 Lines 29−33:
"The model is integrated for seven days from a state of rest, which is sufficient time for the subglacial discharge tracer to reach the southern boundary. Note that the numerical experiment results represent the transitional process of the subglacial discharge tracer, and are not applicable to long-term behavior of subglacial plume. Thus, we do not discuss the response of the fjord stratification to the water mass transformation due to plume entrainment on the timescales of months."

In Page 10 Lines 29−33: We have already described as follows.
"However, this study revealed the transitional processes of subglacial discharge plume over a

relatively short time scale. To fully understand the longer-term interactions of subglacial discharge and fjord stratification (e.g., seasonal and interannual variations), we need to perform long-term oceanic observations and numerical experiments to capture the realistic nature of discharge and submarine melting over a much broader model domain."

**Page 7, Line 42:  A caveat which may be worth noting is that turbidity at the fjord surface may only be a reasonable proxy if the plume is able to reach the surface, which depends on the degree of entrainment.**

According to your suggestion, we have added "It should be noted that turbidity at the fjord surface is only a reasonable proxy if turbid subglacial discharge plume is able to reach the surface, which depends on the degree of entrainment." in Page 8 Lines 34−36.

**Page 8 (throughout): Consider an additional run ST16Q600, which uses the stratifi- cation observed in 2016, with a discharge that is 20 percent higher than the CTRL run, since you state the PDD is 20 percent higher in 2016 compared to 2014, so a more realistic representation of the 2016 state would take into account the increase in discharge as well.**

We have performed the additional run (ST16Q45, which uses the stratification observed in 2016, with a discharge that is higher than CTRL) and added the description on the ST16Q45 run in Page 12 Lines 12−19 (Appendix C).

**Page 8 (sect. 5.2) : Why are the model results presented in the paper's discussion section - aren't these also results?**

This paper mainly focused on the observations and thus the model results were shown only in the discussion section in the previous version. We agree with your comment and have added the section of model results (Please see Sect. 5.2).

**Page 22 (also, see other relevant figures and discussions): Each of these measure- ments represents a snapshot of the turbidity, temp, salinity etc., but are these snap- shots representative of the entire 2014 and 2016 melt seasons? How much variability in the discharge rate, turbidity etc. would we expect within a single melt season? I'm not sure whether it's possible to even distinguish differences between the 2014 and 2016 melt seasons from a single sample from each season. If not, then attempting to explain them using the model is not a valid approach.**

We do not think that these snapshots are representative of the entire 2014 and 2016 melt seasons throughout the water column. However, the features in the deeper layer of the fjord can be the results

for the entire melt season.

In Fig. R1 (Please see Response to Comment from Referee #1 Why don't you force your model with RCM runoff estimates?....), the estimated daily mean discharge rate varied largely ($19 \pm 16$ m$^3$ s$^{-1}$ in 2014 and $21 \pm 18$ m$^3$ s$^{-1}$ in 2016). Moreover, the vertical distribution of turbidity, temperature, and salinity in the shallow layer that plume water spreads can be strongly affected by the short-term change in discharge rate. Thus, these vertical distribution in the shallow layer varied at short time scales. In this study, observational results on the turbid subglacial plume water and numerical model results represent the short-term transitional process of subglacial discharge plume. We have added the caveats in Page 8 Lines 42–43 as follows. "It should be noted that the observational data represent snapshots in summer, and not the entire melt seasons. Thus, we discuss only the difference in the short-term transitional process of outflowing plume water."

On the other hand, as shown in Sect. 6.3, submarine melting layer was developed in 2014 and seasonal pycnocline in 2016. These processes could be influenced by the thick warm AW layer and the enhancement of winter vertical mixing, and thus are characterized as the long-term process (annual). Therefore, we think that it is valid to consider the difference in these developed layer (submarine melting and seasonal pycnocline) as the interannual difference. We have already described in Page 2 Lines 21–23 as follows. "The properties of the water masses can change on various time scales from intra-seasonal to interannual or longer-term, reflecting the variabilities on much larger spatial scales."

To fully link the multi-year observations of ocean environments and realistic subglacial discharge scenario are far beyond the capability of the present data and computational resource. We hope this attempt to be the first step toward such kind of observation and modeling.

**Technical Corrections:**

**Page 1, Line 1 and many others. When "structures" is used to denote the properties of vertical water mass profiles i.e. "water structures", this should instead be "vertical density profiles" or equivalent for clarity. For instance, consider changing the title to: "The effect of subglacial discharge on vertical density profiles in Bowdoin Fjord, north- western Greenland."**

We have changed the title "Water mass structure and the effect of subglacial discharge in Bowdoin Fjord, northwestern Greenland" to "Vertical distribution of water mass properties under the influence of subglacial discharge in Bowdoin Fjord, northwestern Greenland" in Page 1 Lines 1–2 and changed "water mass structure" to the other words (e.g., properties of vertical water mass profiles) through the manuscript.

**Page 1, Line 23: Consider rewording the last sentence of the abstract for clarity e.g.: "Fjord**

**stratification is an important factor controlling the vertical distribution of freshwater outflow due to subglacial discharge strength and entrainment. The fjord stratification does not influence the magnitude of subglacial discharge 'amount', as is implied in the original statement.**

We have changed the last sentence "This study indicates that ambient fjord stratification difference is an important factor controlling the vertical distribution of subglacial discharge, together with its amount." to "This study indicates that both fjord stratification and the amount of discharge are important factors in controlling the vertical distribution of freshwater outflow." in Page 1 Lines 22–23.

**Page 2, Line 34: "...of a proglacial fjord." Page 2, Line 42: Correct to present tense: "This study focuses on BF ...."**

Corrected. Please see Page 2 Line 39 and Page 3 Line 4.

**Page 3, Line 1: Correct to: "In June and July, the sea ice melts rapidly exposing the open ocean surface."**

Corrected. Please see Page 3 Line 6.

**Page 3, Line 6. If the depth of PW/AW interface is between 50m and 150m, then 210m should always be below the AW and PW interface. In Fig. 12, the schematic shows a PW/AW interface that is between 175m and 225m.**

This sentence represents the depth of PW and AW cores. PW/AW interface is between around 210m as shown in Figure 12. To clarify, we have changed "corresponds to the depth between warm AW (at the deepest part of the fjord) and cold PW (depth: 50–150 m)" to "corresponds to the depth between warm AW and cold PW cores" in Page 3 Line 11.

**Page 3, Line 24: "0.01" psu? Page 3, Line 36: "...which subsequently spreads due to entrainment."**

Corrected. Please see Page 3 Line 36 and Page 4 Lines 5–6.

**Page 4, Line 9 and others: Consider another word instead of "endmember" which is unclear when it is used... perhaps "source?"**

We have changed "endmember" to "source" through the manuscript.

**Page 5, Line 27 and others: Consider a more compact notation $\theta_{\max}^{2014}$, etc.**

We have changed $\theta_{\max2014}$ and $\theta_{\max2016}$ to $\theta_{\max}^{2014}$ and $\theta_{\max}^{2016}$, respectively.

**Page 9, Line 29:** As a point of clarification, the subglacial discharge that exits near the surface as in this study would lighten the surface layers and increase strati- fication, but if it preferentially lightens intermediate/deeper layers, it would act to decrease stratification as is observed in other studies (see Jackson et al., 2017: https://doi.org/10.1002/2017GL073602).

We have changed the sentence "In general, fjord stratification is expected to be stronger after subglacial discharges into the fjord, because a density difference is generated." to "Fjord stratification is expected to be stronger after subglacial discharge that exits near the surface as in this study, because subglacial discharge would lighten the surface layers." in Page 10 Lines 28–29.

**Page 9, Line 43:** "In turn, a prominent local ... was detected around 60 m in 2016, which was significantly warmer than the local maximum in 2014."

Corrected. Please see Page 10 Line 42.

**Page 16, Line 4:** Correct to: "... (blue in 2014, red in 2016, both inside Bowdoin Fjord, and green in 2016, outside Bowdoin Fjord)."

Corrected. Please see Page 18 Lines 4–5.

**Fig.12 and to a lesser extent, Fig. 13:** Consider esthetic improvements that would im- prove this figure including fewers arrows, clearer color/font contrast, labels, etc. Is the PW layer here meant to only represent the PW core? It may be clearer to differentiate between $\Theta$min and the full PW layer. It is not clear exactly which portion of the vertical column has a stronger vs. weaker stratification.

We have changed figures 12 and A1 as follows.

・Figure 12:

We have used fewer arrows.

We have used color gradient between PW and AW cores.

We have represented the strength of stratification clearer.

・Figure A1:

We have removed arrows.

We have changed the color of line indicating the submarine meltwater fraction.

[revised manuscript text omitted]

---

## Author Comment (AC2) · 9 Dec 2019

We thank you for your comments. Please see the attached supplement for our responses, new manuscript, table, and figures.

Please also note the supplement to this comment:
https://www.ocean-sci-discuss.net/os-2019-33/os-2019-33-AC2-supplement.pdf

———————————————

---

## Author Response (AR2)

Revision of the manuscript [Paper #os-2019-33] "Vertical distribution of water mass properties under the influence of subglacial discharge in Bowdoin Fjord, northwestern Greenland" by Ohashi et al.

We thank Dr. Mario Hoppema for careful reading of our manuscript and for giving useful comments and suggestions. We have addressed the raised your comments and explained how we revised.

Your comments are written in bold and replies from the authors follow your comments. Changes are highlighted in the manuscript using the red color.
* * *
**Thank you for the resubmission of your manuscript. I am generally satisfied with the changes you made as a response to the reviews. Below my final comments and suggestions are listed. Please prepare your final submission for publication in Ocean Science.**

Thank you very much for your positive assessment and constructive comments. We have responded to all comments.

**P1, L21 suggests instead of indicates**

We have changed "indicates" to "suggests" in P1 L21.

**P1, L40 "The plumes have higher fraction of subglacial discharge than those in the deep fjords due to greater entrainment of ambient water." I am a little confused by this sentence. I think you mean "The plumes of shallow fjords have a higher fraction of subglacial discharge", but this cannot be caused by the greater entrainment of ambient water; a greater entrainment of ambient water would tend to reduce the fraction of subglacial discharge, right? Please explain and correct, if necessary.**

This sentence was misreading. We have changed "greater" to "less" in P1 L41.

**P2, L24 temporally instead of temporarily?**

We have changed "temporarily" to "temporally" in P2 L24.

**P2, L25 I think you mean: the impact of varying fjord stratification on the vertical distribution of outflowing plume water … ; or something similar. As it is now, it is not clear.**

We have changed this sentence to "the impact of varying fjord stratification on the vertical distribution of outflowing plume water …" in P2 L25–26.

**P2, L34-35 This sentence says exactly the same as the previous one. It should be deleted.**

We have deleted this sentence in P2 L34–35.

**P3, L4 delete: exposing open ocean surface. It is self-evident that after melting there is open ocean water.**

We have deleted this part "exposing open ocean surface" in P3 L4.

**P3, L23 I think the notation 14D1-6 and 16D1-6 for all stations is not clear. Please modify, for example, to 14D1 to 14D6**

We have changed "14D1–6" to "14D1–14D6" and "16D1–6" to "16D1–16D6", respectively in P3 L23.

**P3, L27, L32 (and at other places in the text) Please do not use psu as a unit for salinity. This is not a unit. Salinity is a ratio and thus without physical unit. Alternatively, please indicate the salinity scale, for example Practical Salinity Scale.**

We have changed "psu" to "practical salinity scale (PSS)" throughout the text.

**P4, L27 specifically instead of significantly?**

We have changed "significantly" to "specifically" in P4 L27.

**P4, L28 θ–S properties were similar …**

Corrected. Please see P4 L28.

**P4, L36 delete: "The vertical structures of temperature were notably different between 2014 and 2016." This info was already given earlier and again after this sentence.**

We have deleted this sentence in P4 L36.

**P4, L38-39 "with a temperature of −0.8 °C in both observations." It is not clear which observations are meant. Please modify wording.**

We have changed "in both observations" to "in both 2014 and 2016" in P4 L38–39.

**P5, L13 "Salinity varied vertically between 2014 and 2016." This can be deleted because this info is given in the next sentences.**

We have deleted this sentence in P5 L13.

**P6, L2-3 "The results were similar for the other stations." It is not clear to me which other**

**results are meant here. Please explain and/or modify.**

Other results mean the differences in the freshwater fractions between 2014 and 2016 inside Bowdoin Fjord (e.g., Stations 14D4 and 16D4), except the site nearest the ice front (Stations 14D1 and 16D3). To clarify, we have changed this sentence to "The results were similar for the other stations regardless of distance from the ice front (e.g., Stations 14D4 and 16D4 with ~11 km distance from the ice front)" in P6 L2–4.

**P6, L4-5 "The θ –S properties obtained in 2014 were closer to those of the AW core than the PW core (Figs. 7a and 7c), whereas they were more similar to the PW core" This sentence does not make sense. Please correct.**

We have changed this sentence to "The $\theta$–$S$ properties obtained in 2014 were closer to those of the AW core than the PW core (Figs. 7a and 7c), whereas those in 2016 were closer to the PW core (Figs. 7b and 7d)." in P6 L6.

**P6, L7-9 "In both 2014 and 2016, near the PW core, the θ –S properties deviated slightly from the melt line and were located outside the meltwater quadrant, implying the influence of PW." Especially the last part is not clear. If near the PW core, it is obvious that there is influence from the PW core. Please explain better what you mean.**

This sentence means "In both 2014 and 2016, near the PW core, the $\theta$–$S$ properties deviated slightly from the melt line and were located outside the meltwater quadrant. This implies that there is influence from the PW in both years". We agree with your comment and think that it is obvious that there is influence from the PW near the PW core. Because the last part "implying the influence of PW" is redundant, we have deleted the last part in P6 L10.

**P8, L12 Please modify: In Sect. 4.2, it was shown that high turbidity corresponded …**

We have added "it was shown that" in P8 L13.

**P8, L24-26 " However, because the turbidity distribution at the fjord surface can be visually captured by satellites (Chu et al., 2012) and drones, it is easier to monitor the distribution of turbidity than to investigate the subglacial discharge fraction based on in situ observations." I do not understand the intention of this sentence. It is easier indeed, but only qualitative estimations are possible (much additional info is missing). You just showed that your method can provide quantitative info. So this is about different things.**

**P8, L27 "This emphasizes the importance of turbidity measurements to better understand the subglacial plume behavior." I think this sentence is superfluous, considering all info given before.**

We agree with your comments and have deleted these sentences in P8 L25–28.

**P9, L39-40 "The available excess heat of up to 0.9 °C …" Heat is not the same as temperature. Please be accurate in writing.**

We have deleted "of up to 0.9 °C" in P9 L40–41.

**Figure 1 It is not clear what the location is of station 14 (blue) and station 16 (red) outside the fjord. In green it also says in the right corner of panel (b) St.16D6. What does that mean? Is St.16 the same as D6? Please clarify. I think the notation St.14 and St.16 is misleading. The first thing that comes to mind when reading this is station 14 and station 16, whereas you mean all stations in 2014 and in 2016, respectively. St.16D6 is one station indeed, which is fine as notation. Please change the notations for avoiding confusion.**

These notation (St. 14, St. 16, and St. 16D6) outside the fjord represent figure legends. To clarify, we have removed these notation (St. 14, St. 16, and St. 16D6) and changed the notation of each station in Figure 1b. Please see P17.

**Figure 2 Please do not use psu for salinity on the axes. In the caption: diagram**

We have used PSS in Figure 2 and changed "diagrams" to "diagram" in the caption. Please see

P18.

**Figure 3 The figures in these panels are not vertical profiles (which are diagrams with depth on the y-axis against the concentration). They are called contour plots or (hydrographic/temperature) sections.**

We have changed "vertical profiles" to "contour plots" in P19.

**Figure 7 Please indicate in the figure, or describe in the caption, the meltwater quadrant.**

We have indicated the meltwater quadrant in Figure 7. Please see P23.

**Figure B1 Same as Figure 3: These are not vertical profiles**

[revised manuscript text omitted]